# Biomarkers of pediatric Epstein-Barr virus-associated hemophagocytic lymphohistiocytosis through single-cell transcriptomics

Jie Shen[1,2,3,19], Yunyan He[4,19], Hong Zheng [1,2,3,5], Jianwen Xiao[6], Fu Li[7], Keke Chen[8], Biyun Guo[9], Yulei He[10], Lin Liu[1,2,3], Zhi Lin [1,2,3], Dan Wang[1,2,3], Leping Liu[1,2,3], Shengfeng Wang[1,2,3,11], Wen Zhou [12], Yingchi Zhang[13], Jian Wei[14], Yunchu Wang[15], Rong Hu[16], Daolin Tang [17], Dao Wang[18] ✉ & Minghua Yang [1,2,3] ✉

Epstein-Barr virus-associated hemophagocytic lymphohistiocytosis (EBV-HLH) is a fatal hyperinflammatory disorder distinct from self-limiting EBV-induced infectious mononucleosis (IM). However, the immunological mechanisms underlying the divergence between benign EBV infection and fulminant HLH—particularly in the absence of inherited immunodeficiency—remains unclear, and systematic comparisons of immune landscapes across EBV-associated disease spectra are lacking. In this study, by enrolling children with IM and healthy volunteers as controls, we utilize single-cell RNA sequencing to identify unique immunological characteristics of EBV-HLH. Our analysis indicates that patients with EBV-HLH exhibit widespread activation of NF-κB signaling pathway. Furthermore, excessive cytokine secretion by T and NK cells is observed, along with a shift in monocyte differentiation towards an inflammatory phenotype, and the aggregation of IDO1[+] monocytes. Metabolic pathway analysis reveals that L-kynurenine, a downstream metabolite of IDO1, is specifically elevated in EBV-HLH and mediates the production of multiple pro-inflammatory cytokines. Collectively, our study maps the immune landscape in pediatric EBV-HLH at single-cell resolution, uncovering potential role of IDO1[+] monocytes and L-kynurenine as biomarkers.

Epstein-Barr virus (EBV), a member of the Herpesvirinae subfamily, is one of the eight major herpesviruses known to cause human diseases[1]. While the majority of individuals infected with EBV remain asymptomatic throughout their lives, symptomatic infections predominantly occur in children, manifesting as infectious mononucleosis (IM) during the acute phase of the infection[2,3]. Although IM typically follows a self-limiting course with minimal complications, acute EBV infection can occasionally precipitate the development of a severe syndrome known as hemophagocytic lymphohistiocytosis (EBV-HLH)[4,5]. EBV-HLH impairs the physical and mental health of children, imposing a substantial burden on affected families. Early detection and timely treatment can improve patient outcomes, making it crucial to explore the pathogenic mechanisms and identify biomarkers for the early diagnosis of pediatric EBV-HLH. Furthermore, because EBV-HLH is the most common subtype of secondary HLH, studying it will also enhance our understanding of the pathogenesis of secondary HLH[6].

Current studies on immune response suggest that upon EBV infection, natural killer (NK) cells and cytotoxic T lymphocytes (CTLs) are activated to target and eliminate the infected cells[2]. In individuals with normal immune function, this process leads to the resolution of clinical symptoms, typically manifesting as benign, self-limiting IM. However, when NK cells and CTLs are unable to effectively eliminate the target cells, continuous antigen stimulation occurs. This persistent stimulation triggers an excessive inflammatory response, known as a cytokine storm, which can culminate in HLH[7–9]. While genetic mutations are known to cause immunodeficiency associated with primary HLH[5], the immune mechanisms underlying the progression from EBV infection to EBV-HLH remain largely unknown[10,11], especially in the absence of inherited immunodeficiency.

Single-cell atlases are essential for elucidating the pathogenesis and identifying biomarkers of diseases. However, current single-cell transcriptome studies related to EBV infection have been limited to a single disease[12–15]. The evolutionary trajectories and immune functions of immune cells across different disease cohorts following EBV infection have not been comprehensively studied. The different clinical manifestations caused by EBV infection may be related to the distinct modes of EBV infection. While EBV predominantly infects B cells in healthy carriers and in cases of IM, a proportion of T cells and NK cells are also infected in EBV-HLH[16–19]. This differential pattern of EBV infection may underlie the intact immune responses observed in IM and the dysregulated immunity characteristic of EBV-HLH. A comprehensive comparative study involving asymptomatic EBV carriers, children with EBV-induced IM, and those with EBV-HLH is essential to address key questions: What are the distinct features of the peripheral immune landscape across these clinical states? Which genes and signaling pathways are preferentially activated in EBV-HLH, the most severe manifestation? And are there specific immunological signatures that uniquely define EBV-HLH?

This study utilizes single-cell transcriptomics to investigate healthy children (asymptomatic EBV carriers), children with EBV-IM, and children with EBV-HLH. We reveal that monocytes in EBV-HLH act as key effector cells in the inflammatory storm and identify a new activated monocyte subpopulation: indoleamine 2,3-dioxygenase 1 (IDO1)+ monocytes. Additionally, L-kynurenine, a product regulated by IDO1, may serve as a pro-inflammatory factor in EBV-HLH. Collectively, our study provides a single-cell transcriptomic resource that advances the understanding of pediatric EBV-HLH pathogenesis and uncovers a potential link between metabolic reprogramming and inflammatory storm, offering a multi-layered target framework for future mechanistic and therapeutic investigations.

## Results

### Patient characteristics
A total of 29 pediatric participants were enrolled in the scRNA-seq analysis of this study, comprising 17 patients with EBV-HLH (HLH group), 9 patients with IM group, and 3 healthy volunteers (HV group). The diagnosis of EBV-IM was based on the presence of at least three of the following six clinical features: fever, cervical lymphadenopathy, pharyngitis or tonsillitis, hepatomegaly, splenomegaly, or eyelid edema, along with detectable EBV DNA in peripheral blood. EBV-HLH was diagnosed according to the HLH-2004 diagnostic criteria[20] in conjunction with EBV DNA positivity in peripheral blood. All samples for single-cell sequencing were collected within 0–3 days after admission (acute phase, prior to treatment).

The median plasma EBV DNA levels were $1.31 \times 10^3$ copies/mL (range: $0.63 \times 10^3$–$2.41 \times 10^3$) in the HV group, $1.50 \times 10^4$ copies/mL (range: $1.05 \times 10^3$–$1.02 \times 10^5$) in the IM group, and markedly elevated in the HLH group at $3.84 \times 10^5$ copies/mL (range: $1.73 \times 10^3$–$9.50 \times 10^6$). Clinically, EBV-HLH patients exhibited hallmark features of HLH, including persistent high-grade fever and cytopenia. Bone marrow hemophagocytosis was observed in 88.2% of cases. In 41.2% cases,

reduced NK cell activity was observed, while all cases exhibited elevated levels of soluble CD25 (sCD25). Among the 17 EBV-HLH patients, 15 (88.2%) achieved complete remission (CR) following treatment, while 2 (11.8%) were classified as non-responders (NR). Analysis of EBV-infected lymphocyte subsets in all 17 EBV-HLH cases revealed diverse infection patterns: 2 patients exhibited EBV infection restricted to B cells, 4 had dual positivity in B and NK cells, and 11 demonstrated EBV presence across T, B, and NK cell subsets.

### Identifying the single-cell transcriptional landscape
We obtained a transcriptome dataset comprising 216,817 cells after excluding low-quality cells. Of these, 25,091 cells (11.57%) were from healthy volunteers (HV), 56,752 cells (26.18%) were from individuals with IM condition, and 134,974 cells (62.25%) were from HLH condition patients. After normalizing for read depth and mitochondrial read counts, we compiled a high-quality, batch-free dataset suitable for comparative analysis. Principal component analysis was then conducted (Fig. 1a). Canonical gene markers were employed to classify the transcriptomes into 10 major cell types (Fig. 1b–d and Supplementary Fig. 1a, b). These cell types include CD4+ T cells, CD8+ T cells, γδ T cells, NK cells, NKT cells, B cells, plasma B cells, monocytes, classical dendritic cells (cDC), and plasmacytoid dendritic cells (pDC).

To compare cell distribution across different groups, we calculated Ro/e values and cell proportions for the 10 major cell types (Fig. 1e and Supplementary Fig. 1c–f). Compared to the HV group, the proportion of CD8+ T cells increased in both the IM and HLH groups, correlating with the activation of CD8+ T cells by the immune system in response to EBV infection. In contrast, γδ T cells, B cells, cDC, and pDC were predominantly enriched in the HV group. Plasma cells were enriched in the IM group but decreased in the HLH group, suggesting a potential weakening of humoral immunity in patients with HLH.

To investigate the pathogenic immune response to EBV-HLH, termed the "inflammatory factor storm," we evaluated the functional module scores of cytokines (Fig. 1f) and inflammatory responses (Fig. 1g) in the major cell types. The UMAP plot indicated that high expression of cytokine-related genes was mainly concentrated in T cells and NK cells, with monocytes being the central cell population mediating inflammatory responses.

Overall, this finding suggests that EBV-HLH exhibits a unique transcriptional profile, characterized by both shared and distinct patterns of immune cell dysfunction when compared to IM and healthy control groups.

### Activation of NF-κB signaling in T cells of HLH
To elucidate the heterogeneity among individual T cell subpopulations across the three clinical states, we re-clustered all T cells from PBMCs and performed unsupervised clustering, yielding a total of 13 clusters (Fig. 2a and Supplementary Fig. 2a). The top 10 differentially expressed genes (DEGs) for each cell cluster are depicted in Fig. 2b and Supplementary Fig. 2b. Based on canonical T cell markers (Fig. 2c and Supplementary Fig. 2c), these clusters were categorized into three CD4+ T cell clusters (CD4), nine CD8+ T cell clusters (CD8), and one γδ T cell cluster (T_C13_GDT_TRDV2; TRDV2, TRGV9, TRGC1, TRDC).

Functional scoring results (Supplementary Fig. 2d) identified three naïve T cell subgroups highly expressing naïve gene features (TCF7, SELL, CCR7, LEF1): T_C01_CD4_CCR7, T_C02_CD4_TCF7, and T_C04_CD8_CCR7 clusters. All CD8+ T cell subgroups, except for T_C04_CD8_CCR7, highly expressed cytotoxic effector molecules (GZMB, NKG7, GNLY, PRF1). Among these, T_C05_CD8_STMN1, T_C06_CD8_MKI67, and T_C07_CD8_UBE2C exhibited high expression of proliferation markers. T_C08_CD8_RGS1 and T_C12_CD8_HAVCR2 also highly expressed cytokine genes (IFNG, TNF, CXCR3, CCL3, CCL4) and exhaustion markers (HAVCR2, CTLA4, LAG3, PDCD1, LAYN), consistent with the functional scoring results (Supplementary Fig. 2d). The T_C03_CD4_MAF cluster had relatively high cytotoxic and exhaustion

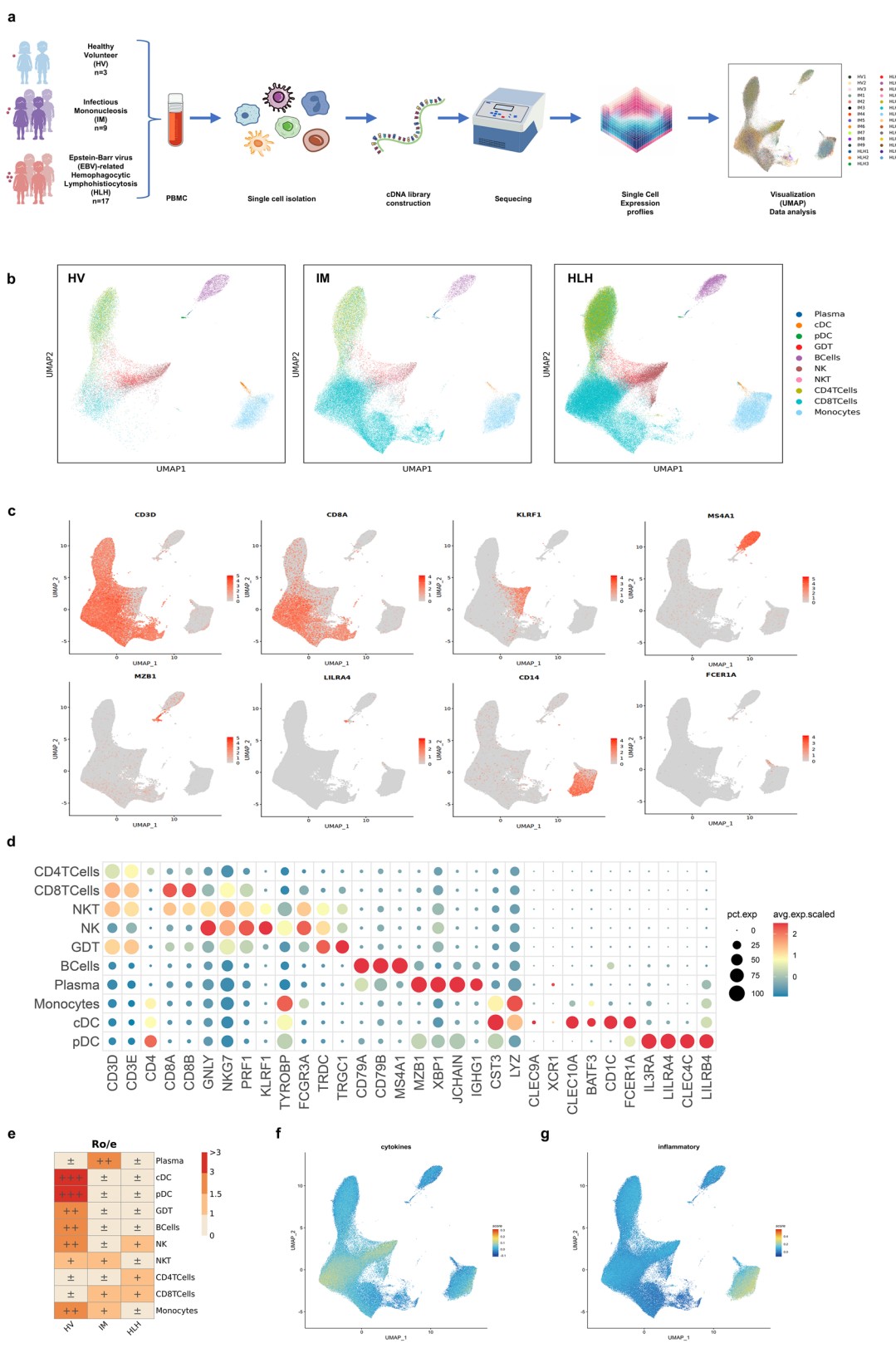

scores (Supplementary Fig. 2d), suggesting these cells might be CD4⁺ CTLs. Additionally, the T_C02_CD4_TCF7 and T_C03_CD4_MAF clusters highly expressed Treg markers (*FOXP3, IL2RA, TNFRSF4*) (Fig. 2c). However, these clusters had a low proportion of positive cells (Supplementary Fig. 2c), which may be due to the mixing of CD4⁺ T cells with a small fraction of Treg cells.

Next, the degree of cellular enrichment was quantified based on the Ro/e value of each cell cluster (Fig. 2d and Supplementary Fig. 2e). CD8⁺ CTL clusters, especially the T_C05_CD8_STMN1, T_C06_CD8_MKI67, and T_C07_CD8_UBE2C, were most enriched in the IM group (Supplementary Fig. 2f, g). In contrast, the T_C08_CD8_RGS1 and T_C12_CD8_HAVCR2 cell clusters, which have

**Fig. 1 | Single-cell atlas of EBV-HLH patients, IM patients, and healthy volunteers. a** Flowchart describing the overall experimental design of this study. **b** UMAP was used for mapping 25,091 single cells from HV ($n = 3$), 56,752 single cells from IM ($n = 9$), and 134,973 single cells from HLH ($n = 17$). Based on the expression levels of marker genes, cells are categorized into ten distinct clusters and color-coded accordingly. **c** Expression distribution of canonical markers identifying individual cell clusters in the dot plot. The diameter of each dot reflected the proportion of subtype cells expressing a specific gene, while the color denotes the normalized mean expression. **d** The main cell lineages were labeled according to cell identity using canonical cell markers, as depicted in the UMAP plot. Color intensity ranging from gray to red indicates expression levels from low to high. **e** The enrichment of each cell subpopulation was quantified by calculating the ratio of observed to expected cell numbers (Ro/e). +++, Ro/e > 3; ++, 1.5 <Ro/e ≤ 3; +, 1 ≤ Ro/e ≤ 1.5; +/−, 0 <Ro/e < 1. **f** The scoring of cytokine sets for each cell in UMAP plot, with color intensity transitioning from blue to red to denote scores from low to high. **g** The scoring of inflammatory response sets for each cell in UMAP plot. The gradient from blue to red signifies expression levels increasing from low to high.

high cytokine and exhaustion scores, were enriched in the HLH group. As expected, the exhaustion score of CD8 + T cells in the HLH group was significantly higher than that in the IM group (Supplementary Fig. 2h). This may be related to the insufficient antiviral capacity and over-activation of cytokine production in T cells in HLH.

To further investigate the differential transcriptomic alterations in T cells (excluding γδ T and NKT cells) within children with HLH, we compared the DEGs of T cells in HLH patients to those in HV or IM patients and conducted enrichment analyses. Compared to HV (Fig. 2e), the upregulated genes in HLH were mainly enriched in processes, such as oxidative phosphorylation, mitochondrial electron transport, cytochrome c to oxygen, cell killing, regulation of canonical NF-κB signal transduction, and positive regulation of cytokine production. In contrast to IM (Fig. 2f), the upregulated genes in HLH were involved in cellular response to cytokine stimulus, leukocyte migration, regulation of apoptotic signaling pathways, and regulation of canonical NF-κB signal transduction.

We further screened for genes specifically upregulated or downregulated in the HLH group. Genes involved in the positive regulation of canonical NF-κB signal transduction (*NFKBIA, S100A12, TNFSF10, TRIM22, TMED4*) and the JAK-STAT signaling pathway (*CISH, PIM1, SOCS3*) were upregulated in the HLH group (Fig. 3a). The activation of NF-κB signaling pathways can lead to increased cytokine production[21]. Consistent with the single-cell sequencing results, flow cytometry analysis confirmed that phosphorylated p65 levels in T cells were elevated in the HLH group compared to the IM group (Fig. 3b), indicating the activation of the canonical NF-κB pathway in EBV-HLH. As expected, most T cell subgroups had the highest cytokine scores in the HLH group (Fig. 3c). Genes involved in the regulation of vesicle-mediated transport, regulation of receptor-mediated endocytosis, and positive regulation of cell adhesion were specifically downregulated in the HLH group (Fig. 3d). The downregulation of these functions could be a potential mechanism leading to the impaired immune response of T cells in HLH.

HLH samples were further stratified into two subgroups based on clinical testing for EBV infection in T cells: the EBV-T group (patients with EBV-infected T cells) and the non-EBV-T group (patients without EBV-infected T cells) (Fig. 3e). Comparative transcriptomic analysis revealed substantial functional heterogeneity in T cell gene expression profiles between these two groups. In the EBV-T group, upregulated genes were predominantly enriched in innate immune-related pathways, including antiviral responses, type I interferon signaling, ribosome assembly, and cytoplasmic translation. Additionally, pathways involved in leukocyte migration, cytokine production, and NF-κB signaling were significantly activated. In contrast, downregulated genes in the EBV-T group were associated with adaptive immune processes such as T cell activation, antigen receptor signaling, cytotoxicity, antigen presentation, and lymphocyte proliferation. The concurrent activation of innate immune responses and suppression of adaptive immunity in EBV-infected T cells reflects a paradoxical immune state that may underlie the susceptibility to cytokine storm in a subset of EBV-HLH patients.

*SH2D1A, ADA*, and *RAB27A* are genes known to be involved in the pathogenesis of HLH[4,22–24]. These genes were highly expressed in the IM group but downregulated in the EBV-T group (Fig. 3f). This pattern suggests that direct EBV infection of T cells may suppress the expression of key genes involved in T cell activation (*SH2D1A*), purine metabolism (*ADA*), and antigen processing or presentation (*RAB27A, SLAMF7*), thereby impairing antiviral immune responses and facilitating viral persistence. In contrast, expression levels of these genes in the non-EBV-T group closely resembled those observed in the IM group. Notably, *CD81* expression followed a graded pattern, with high expression in the IM group, intermediate levels in the non-EBV-T group, and markedly reduced expression in the EBV-T group. As a tetraspanin involved in immune synapse formation, cell adhesion, and signal transduction[25], reduced *CD81* expression might compromise T cell activation and regulation, thereby promoting uncontrolled inflammation. Furthermore, EBV infection could exacerbate this effect by further suppressing *CD81* expression in T cells.

In the HLH-D (the deceased HLH sample) group, the upregulated genes were primarily enriched in NF-κB signaling activation, cytokine storm, and intrinsic apoptosis pathways (Fig. 3g). Conversely, the downregulated genes were mainly enriched in oxidative phosphorylation, positive regulation of leukocyte activation, regulation of hematopoiesis, and positive regulation of binding, suggesting impaired metabolic function and dysregulated T cell activation in the HLH-D group.

In addition, compared to the HV group, *ANXA1* expression was elevated in the IM group compared to healthy volunteers (HV), but notably downregulated in the HLH group (Fig. 3d), with further reduction observed in the HLH-D subgroup (Fig. 3h). These findings suggest that, as an endogenous anti-inflammatory molecule[26–28], *ANXA1* expression levels may correlate with disease severity and prognosis.

## Activation of classical inflammatory signaling in NK cells of HLH

All NK cells (excluding NKT cells) were re-clustered, resulting in five distinct clusters (Fig. 4a and Supplementary Fig. 3a). The top 10 DEGs for each cluster are presented in Fig. 4b and Supplementary Fig. 3b. NK cells are traditionally categorized into two main groups based on CD56 (*NCAM1*) and CD16 (*FCGR3A*) expression levels: CD56[bright]CD16[low] and CD56[dim]CD16[high] [29]. Based on functional gene expression (Fig. 4c), we identified one CD56[bright]CD16[low] cell cluster (NK_C03_XCL2) and two CD56[dim]CD16[high] cell clusters (NK_C01_GZMH and NK_C02_AKR1C3). The remaining two clusters (NK_C04_STMN1 and NK_C05_MKI67) exhibited high proliferative activity. Consistent with previous studies[29], CD56[bright]CD16[low] C03_XCL2 NK cells highly expressed cytokine-related genes, such as *LTB* and *IL2RA*. The two CD56[dim]CD16[high] NK cell clusters highly expressed cytotoxic effector genes, including *PRF1* and several granzymes (*GZMB, GZMA, GZMH*), except for *GZMK*, which was expressed only in CD56[bright]CD16[low] C03_XCL2 NK cells. CD56[dim]CD16[high] C01_GZMH NK cells with high expression of *KLRC2* (NKG2C) were identified as adaptive NK cells.

Compared to healthy volunteers (Fig. 4d), the upregulated DEGs in HLH were primarily involved in oxidative phosphorylation, response to type II interferon, cell killing, and positive regulation of the inflammatory response. In contrast to the IM group (Fig. 4e), the upregulated genes in HLH were enriched in pathways including cell activation, granulocyte chemotaxis, granzyme-mediated programmed cell death signaling, cellular response to cytokine stimulus, and

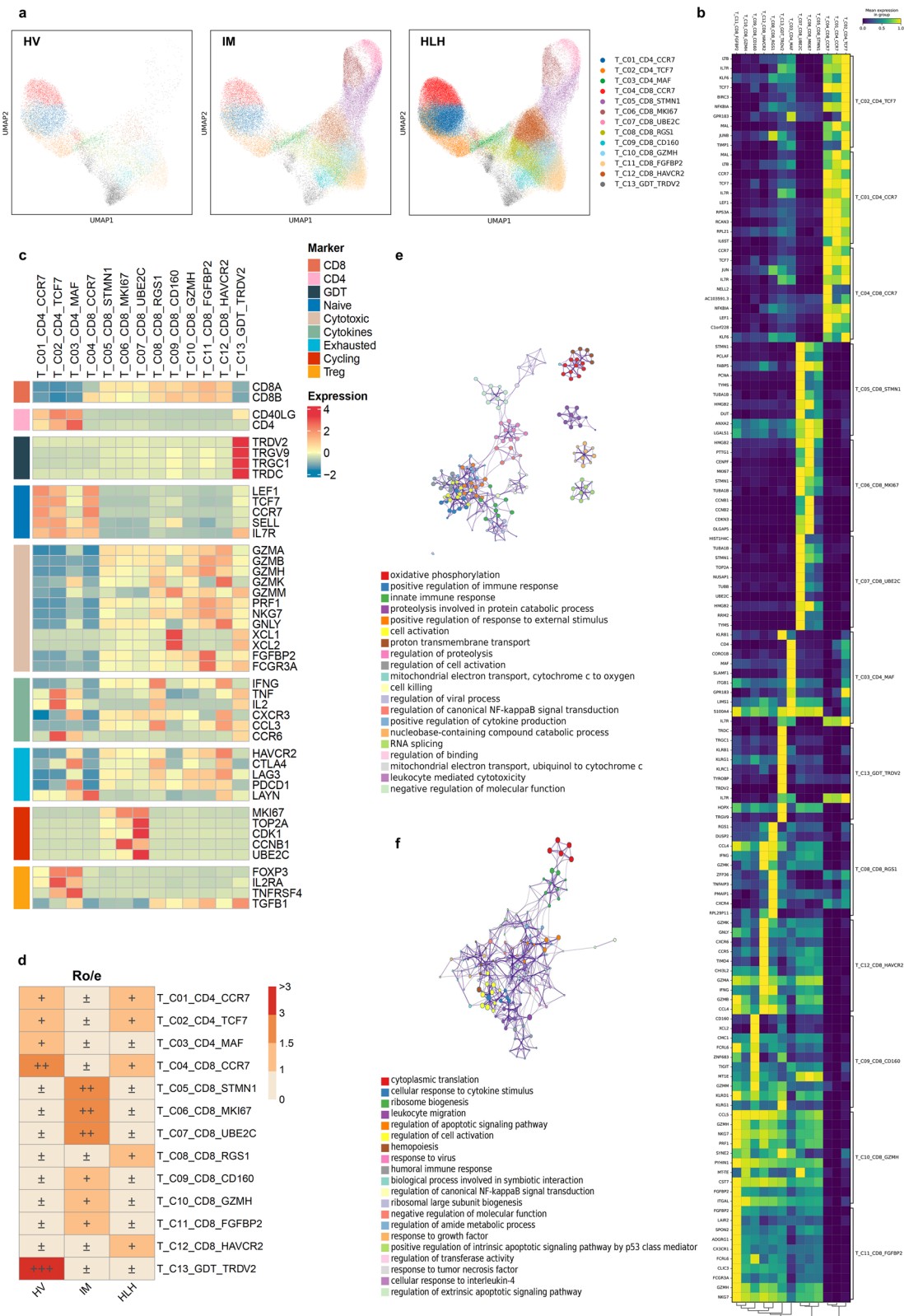

**Fig. 2 | Sub-clustering analysis on T cells. a** UMAP visualization of 13,163 single cells from HV (*n* = 3), 44,746 single cells from IM (*n* = 9), and 107,388 single cells from HLH (*n* = 17). Cells are color-coded and divided into 13 clusters. **b** Heatmap illustrating the average expression values of the top 10 highly expressed genes within each T cluster. **c** The average expression of selected markers of T cells in Heatmap. **d** Quantification of cell subpopulation enrichment based on Ro/e for cellular populations. **e**, **f** Enriched GO terms for upregulated genes in HLH patients compared to HV (**e**) and IM (**f**). A selection of indicative terms from the complete cluster has been transformed into a network configuration. Each term is depicted as a circular node, with the node's size corresponding to the quantity of input genes associated with that term, and its color indicating its cluster membership (i.e., nodes sharing the same color are part of the same cluster). Terms that exhibit a similarity score greater than 0.3 are interconnected by an edge, where the edge's thickness denotes the score of similarity.

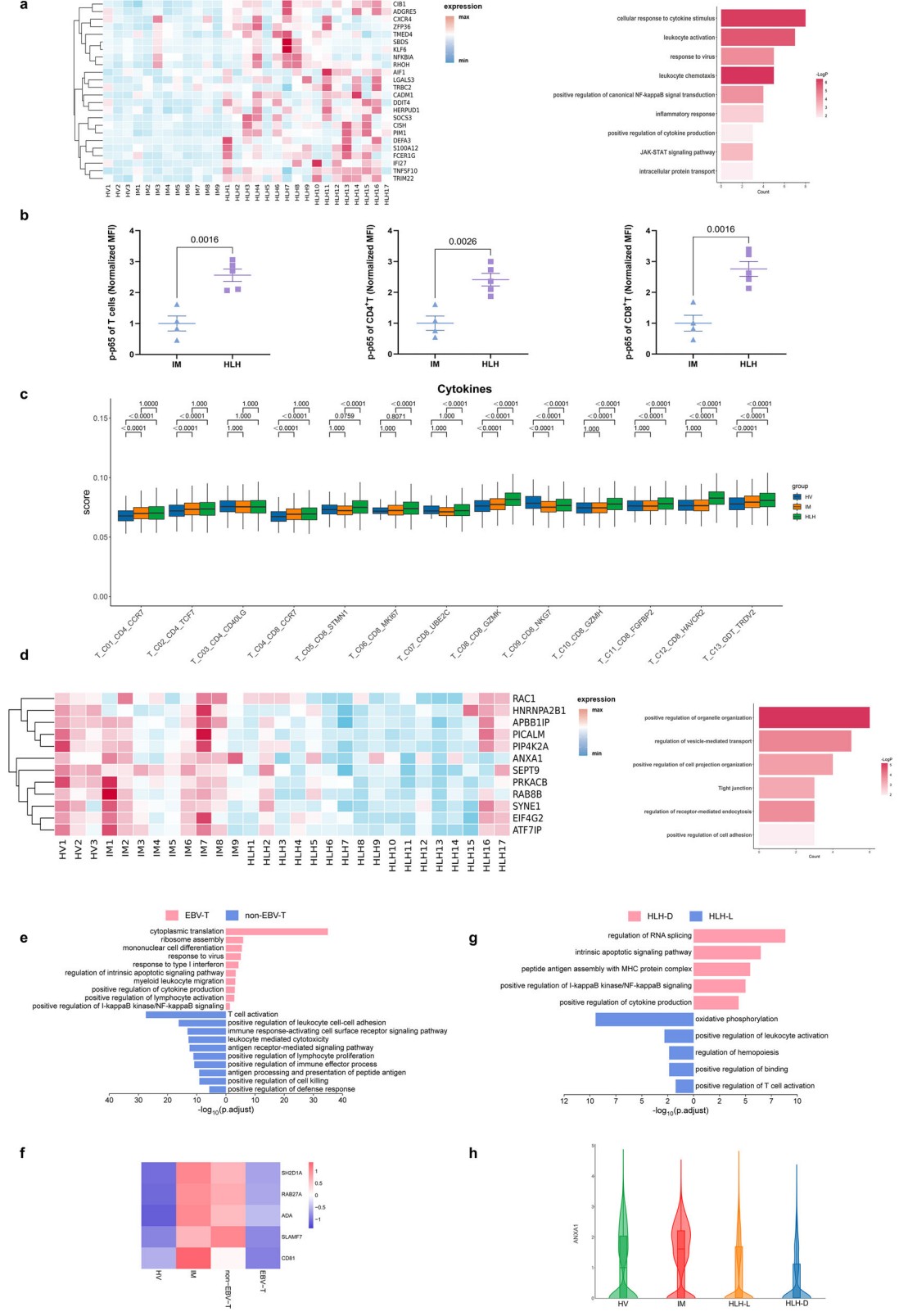

regulation of apoptotic signaling. These findings support that HLH involves different pathway activation.

A more detailed analysis identified genes specifically upregulated or downregulated in the HLH group. Markers of reactive oxygen species and oxidative stress (*GSTP1, LDHA, HSPB1, PRDX5, PGD*) were abnormally upregulated in HLH (Fig. 5a), indicating that NK cells in HLH patients might be experiencing increased oxidative stress.

Similarly, genes associated with the canonical NF-κB pathway were upregulated, and this was corroborated in clinical samples by elevated phosphorylation levels of p65 in the HLH group compared to the IM group (Fig. 5b).

Among the downregulated genes (Fig. 5c), we identified *RASGRP1*, a gene previously associated with EBV-HLH[30]. Genes involved in processes, such as positive regulation of leukocyte cell-cell adhesion,

**Fig. 3 | Immunological characterization of T cell clusters. a** Based on the screened genes specifically upregulated in the HLH group compared to the HV and IM groups, the expression levels of related genes across all samples and their enrichment in the GO and KEGG pathways. The enrichment results were produced by Metascape and BH-corrected. **b** Quantification of phosphorylated p65 levels (normalized MFI values) in T lymphocytes from pediatric IM ($n = 4$) and HLH ($n = 5$) biologically independent samples. Two-sided Independent-samples $t$-test was applied. **c** Box plot showing the scoring of cytokine sets for each cell cluster among the three groups. Comparisons were made using the two-sided Wilcoxon test and $P$-values were adjusted using the BH correction. **d** Based on the screened genes specifically downregulated in the HLH group compared to the HV and IM groups, the expression levels of related genes across all samples and their enrichment in the GO and KEGG pathways. The enrichment results were produced by Metascape and BH-corrected. **e** Enriched GO terms of differentially expressed genes between EBV-T and non-EBV-T groups. The enrichment results were produced by clusterProfiler and BH-corrected. **f** The normalized expression levels of selected genes in T cells across HV, IM, non-EBV-T, and EBV-T groups. **g** Enriched GO terms of differentially expressed genes between HLH-D and HLH-L groups. The enrichment results were produced by clusterProfiler and BH-corrected. **h** Expression levels of *ANXA1* in T cells across HV, IM, HLH-L, and HLH-D groups.

defense response, regulation of cell killing, and immunological synapse formation (*CD81, CD160, CD6, DOCK8, ITGA4*), were also downregulated, potentially impairing binding between NK cells and target cells and weakening NK cell-mediated immune surveillance and cytotoxic functions. Additionally, compared to the HV group, *DUSP1* was upregulated in the IM group, but downregulated in HLH. *DUSP1* downregulation can lead to sustained activation of the MAPK pathway[31], mediating a more intense or prolonged inflammatory response[32]. Other genes involved in the negative regulation of the MAPK cascade, including *DUSP2, NCOR1, STK38, PDCD4*, and *ITCH*, were also downregulated in the HLH group.

The HLH group was further stratified into EBV-NK (patients with EBV-infected NK cells) and non-EBV-NK (patients without EBV-infected NK cells) subgroups, based on clinical testing for EBV infection in NK cells. Comparative analysis revealed that EBV infection altered both metabolic and immunological profiles of NK cells between the two groups (Fig. 5d). NK cells in the EBV-NK group exhibited signs of metabolic reprogramming, characterized by upregulation of genes involved in ATP metabolism and oxidative phosphorylation. This metabolic activation was accompanied by enhanced expression of genes related to cytotoxicity, cytokine production, and positive regulation of immune defense. However, this activated state coexisted with features of immune dysfunction. Specifically, genes associated with negative regulation of NK cell-mediated cytotoxicity, immune suppression, and apoptotic signaling were also enriched, implying that EBV may facilitate immune evasion by functionally impairing NK cells. Downregulated genes in the EBV-NK group were enriched in pathways related to lymphocyte differentiation, antigen presentation, cellular activation, and cytotoxic effector function.

Furthermore, *LYST*, a known HLH-associated gene[4,5], exhibited comparable expression in HV, IM, and non-EBV-NK groups but was downregulated in the EBV-NK group (Fig. 5e). Similarly, *DUSP1*, the negative regulator of MAPK signaling, was upregulated in the IM and non-EBV-NK groups but downregulated in the EBV-NK group (Fig. 5f). *DUSP1* expression was significantly lower in the HLH-D group compared to the HLH-L group (Fig. 5g), and markedly higher in the IM group. These patterns suggest that reduced *DUSP1* expression may contribute to sustained MAPK activation and could be associated with disease severity and poor prognosis in EBV-HLH.

The NF-κB and MAPK pathways are key regulators of inflammatory responses, and their dysregulation is closely associated with excessive cytokine release in hyperinflammatory conditions[33,34]. In EBV infection, the latent membrane protein LMP1 activates NF-κB through a TRAF2/5-dependent mechanism, suppresses SH2D1A (SAP) expression, and promotes secretion of Th1-type cytokines such as TNF-α and IFN-γ[35]. Concurrently, EBV triggers the MAPK/p38 pathway to promote the production of IL-6 and IL-8[36]. These findings highlight the therapeutic potential of targeting NF-κB and MAPK signaling to mitigate cytokine storms in EBV-HLH (Fig. 5h) and related hyperinflammatory syndromes.

### Reduced memory B and plasma cells in HLH

We re-clustered all B cells and performed unsupervised dimensionality reduction clustering, resulting in five distinct subpopulations (Supplementary Fig. 4a). Each cell cluster is characterized by the top 10 DEGs (Supplementary Fig. 4b). B_C01_TCL1A and B_C02_IGLC2 highly expressed naive B cell marker genes (*IGHD, CR2, CCR7, FCER2*), whereas B_C03_AIM2 and B_C04_CD1C exhibited memory B cell features (*CD27, CD1C*). B_C05_FCRL3 aligned with activated naive B cell characteristics, showing high expression of *IGHD, ITGAX, TBX21, FCRL5*, and *CD19*, but low expression of *CD24, CD38, CD27*, and *CR2*. Additionally, B_C01_TCL1A, B_C02_IGLC2, and B_C05_FCRL3 also highly expressed interferon-stimulated genes (ISGs) (Supplementary Fig. 4c). The composition of B cells in HV and IM groups was similar; however, in HLH, the proportion of memory B cells was reduced, and plasma cells were absent (Supplementary Fig. 4d). This alteration may weaken the immune clearance capability against EBV and other pathogens in HLH patients.

Compared to HV, the upregulated DEGs in HLH were involved in various biological processes, including response to viruses, positive regulation of cytokine production, regulation of type I interferon production, regulation of canonical NF-κB signal transduction, response to type II interferon, inflammatory response, and apoptotic signaling pathways (Supplementary Fig. 4e). Compared to IM (Supplementary Fig. 4f), the upregulated genes in HLH were mainly enriched in processes such as negative regulation of the immune system, response to type II interferon, positive regulation of cytokine production, regulation of canonical NF-κB signal transduction, positive regulation of programmed cell death, negative regulation of cell population proliferation, regulation of IL4 production, and regulation of type I interferon production.

Further analysis identified genes specifically upregulated or downregulated in the HLH group. Genes involved in the positive regulation of canonical NF-κB signal transduction (*BST2, CCR7, S100A12, TNFSF10, TRIM22, PIM2, SHISA5*) and in the negative regulation of cell population proliferation (*B2M, GSTP1, IFIT3, TYROBP, H2AC6, IFITM1, NMI, PIM2, EAF2*) were upregulated in the HLH group (Supplementary Fig. 4g). As expected, we observed elevated levels of p65 phosphorylation in B cells from the HLH group, indicating a higher degree of NF-κB pathway activation compared to the IM group (Supplementary Fig. 4h). Meanwhile, the *SYK* gene, involved in leukocyte differentiation, B cell receptor signaling pathway, and positive regulation of cell adhesion, was downregulated in the HLH group (Supplementary Fig. 4i). SYK encodes a tyrosine kinase, a critical component in BCR activation[37]. Thus, the downregulation of SYK could impede BCR signal transduction, potentially limiting B cell activation and function, and reducing the body's antiviral capability.

### Abnormally enhanced MHC class I antigen presentation of DCs in HLH

The proportion of cDC in the HLH group was reduced compared to the HV and IM groups (Supplementary Fig. 1f). Differential gene expression analysis of cDC between HLH and HV or IM groups, followed by enrichment analysis (Supplementary Fig. 5a, b), showed that genes upregulated in HLH versus HV were primarily associated with antiviral defense and innate immune activation pathways, including viral response, interferon-gamma response, cytokine production, and MHC class I presentation. This indicates that residual cDC in HLH are in a

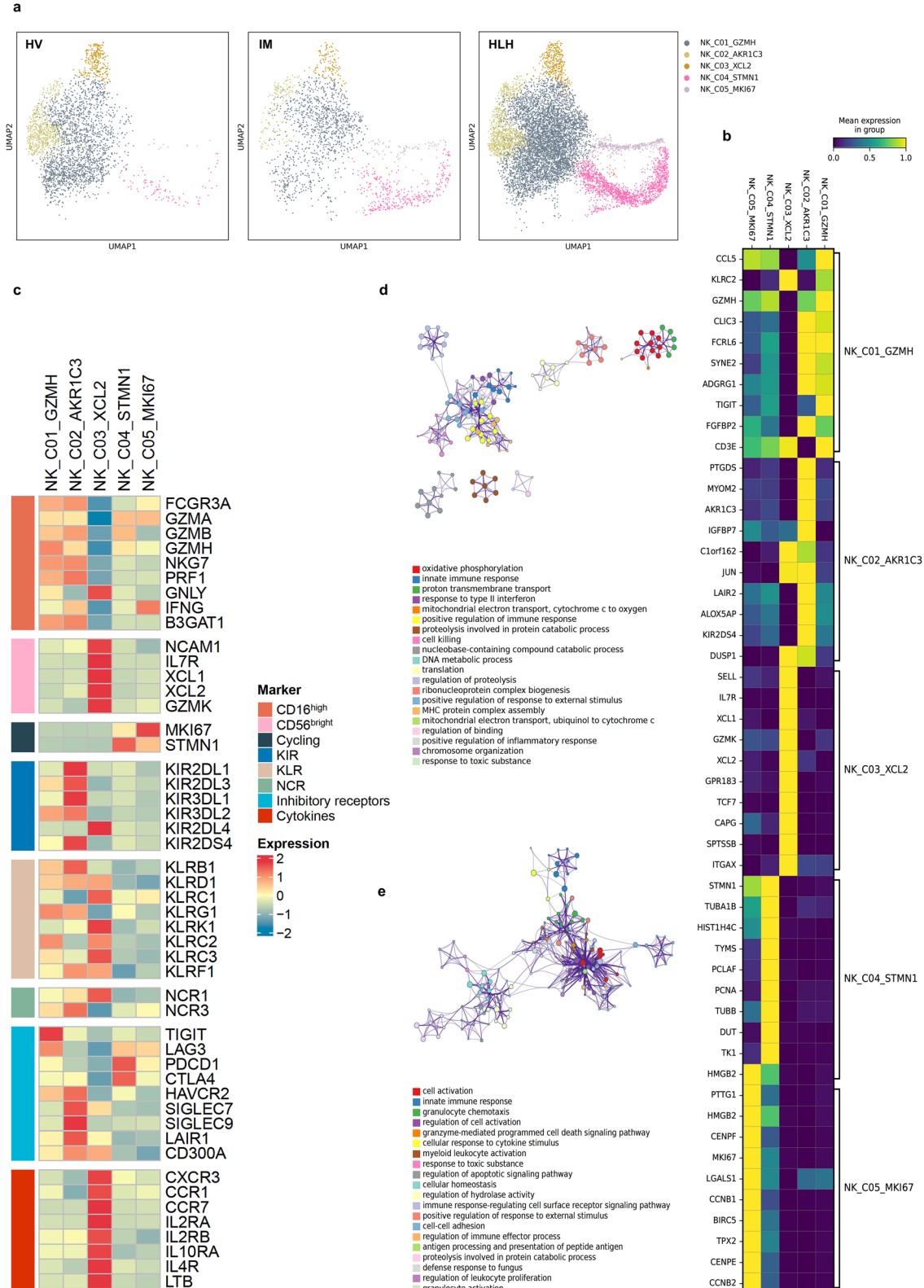

**Fig. 4 | Sub-clustering analysis on NK cells. a** Visualization through UMAP of 2868 single cells from HV (*n* = 3), 1732 single cells from IM (*n* = 9), and 8911 single cells from HLH (*n* = 17). Cells were sorted into 5 clusters, each marked with a unique color code. **b, c** Heatmap representing the average expression values of the highly expressed 10 genes (**b**) and functional genes (**c**) in each NK cluster. **d, e** Enriched GO terms for upregulated genes in HLH patients compared to HV (**d**) and IM (**e**).

highly activated state during viral infection and may drive CD8[+] T cell activation via MHC class I pathways. Concurrently, the upregulation of apoptotic signaling pathways might partially explain the reduced cDC numbers.

In contrast, downregulated genes in HLH versus HV were enriched in pathways related to cytoplasmic translation, ribosome assembly, MHC class II antigen presentation, and cellular homeostasis. When comparing HLH to IM, cDC in HLH showed upregulation of genes

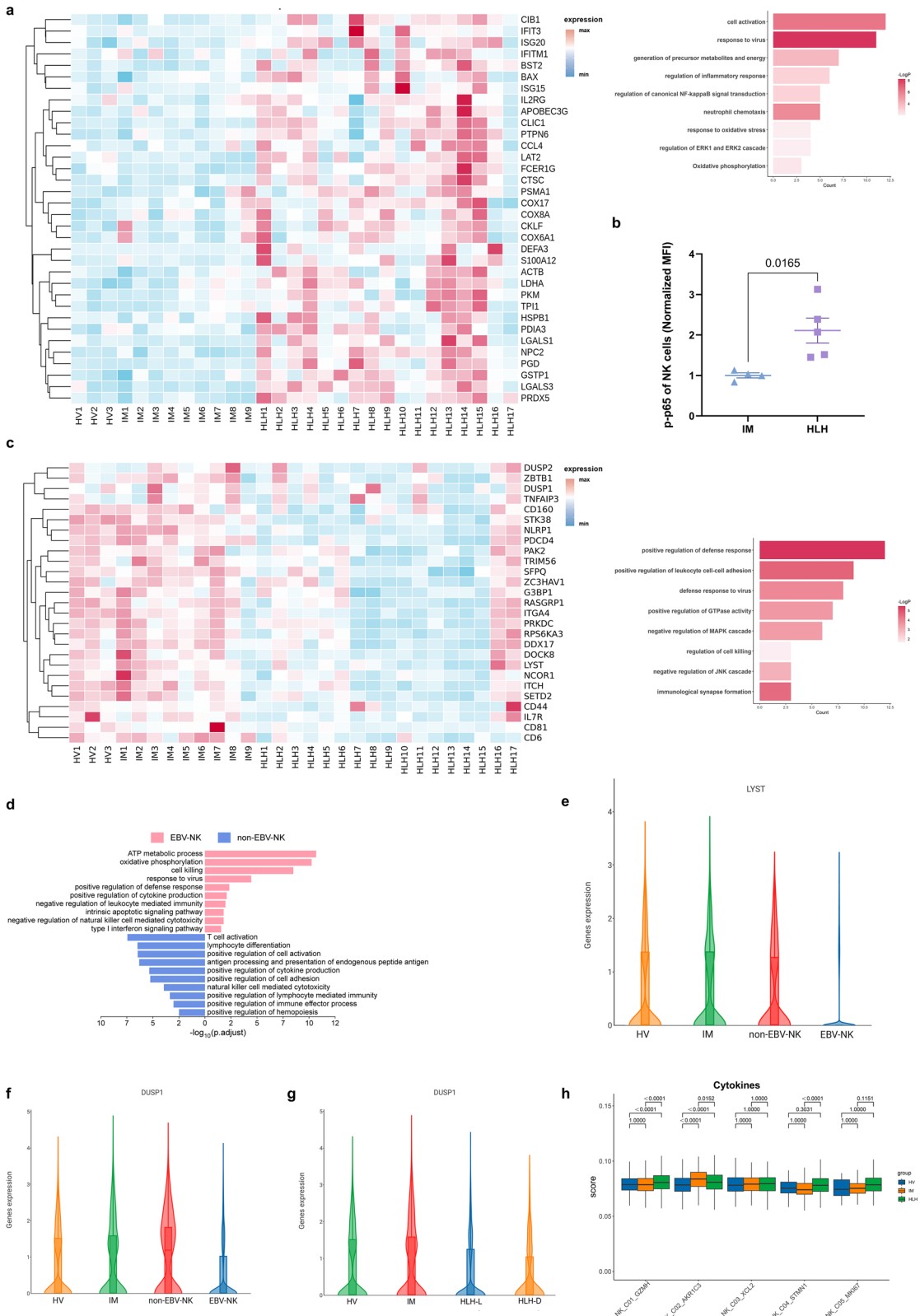

involved in antigen presentation, while downregulated genes were enriched in defense response, cytokine production, and energy metabolism. Additionally, TNF production in cDC was lower in HLH compared to IM. These transcriptional changes suggest alterations in both activation and metabolic profiles of cDC in the HLH group.

pDC were consistently reduced in both IM and HLH groups. As primary producers of type I interferons (IFN-I), pDC depletion likely impairs antiviral immunity. Comparative analysis of DEGs revealed that, in HLH versus HV, upregulated genes in pDC were enriched in antigen processing and presentation (particularly via MHC class I), IFN-γ response, oxidative phosphorylation, ATP metabolism, and T cell activation (Supplementary Fig. 5c). These findings indicate that residual pDC in HLH are metabolically active and engaged in cell-mediated immune responses, potentially contributing to CD8⁺ T cell hyperactivation.

**Fig. 5 | Immunological characterization of NK cell clusters. a** Based on the screened genes specifically upregulated in the HLH group compared to the HV and IM groups, the expression levels of related genes across all samples and their enrichment in the GO and KEGG pathways. The enrichment results were produced by Metascape and BH-corrected. **b** Quantification of phosphorylated p65 levels (normalized MFI values) in NK cells from pediatric IM (n = 4) and HLH (n = 5) biologically independent samples. Two-sided Independent-samples *t*-est was applied. **c** Based on the screened genes specifically downregulated in the HLH group compared to the HV and IM groups, the expression levels of related genes across all samples and their enrichment in the GO and KEGG pathways. The enrichment results were produced by Metascape and BH-corrected. **d** Enriched GO terms of differentially expressed genes between EBV-NK and non-EBV-NK groups. The enrichment results were produced by clusterProfiler and BH-corrected. **e**, **f** Expression levels of *LYST* and *DUSP1* in NK cells across HV, IM, non-EBV-NK, and EBV-NK groups. **g** Expression levels of *DUSP1* in NK cells across HV, IM, HLH-L, and HLH-D groups. **h** Box plot showing the scoring of cytokine sets for each cell cluster among the three groups. Comparisons were made using the two-sided Wilcoxon tests and *P*-values were adjusted using the BH correction.

Similarly, in HLH versus IM, upregulated genes were also associated with MHC class I antigen presentation and cytotoxic T cell regulation (Supplementary Fig. 5d). In contrast, genes downregulated in HLH versus HV were linked to histone modification, myeloid differentiation, IFN-β production, and dendritic cell development, suggesting impaired pDC maturation and reduced antiviral signaling.

In summary, DCs in HLH display enhanced MHC class I antigen presentation, promoting CD8[+] T cell overactivation, while concurrent suppression of protein synthesis and activation of apoptotic programs may drive DCs exhaustion, further compromising immune regulation.

## Identifying monocytes as key effector cells in the inflammatory storm

We also re-clustered all monocytes from PBMCs and performed unsupervised dimensionality reduction clustering, resulting in 10 subpopulations (Fig. 6a and Supplementary Fig. 6a). Each cell cluster was characterized by the top 10 ranked genes in expression levels (Fig. 6b and Supplementary Fig. 6b). Based on the expression levels of CD14 and CD16 (*FCGR3A*) (Fig. 6c and Supplementary Fig. 6c), seven subpopulations (C01 - C07) were identified as classical monocytes (CD14[+], FCGR3A[−]); Mono_C10_FCGR3A as non-classical monocytes (CD14[+], FCGR3A[++]); and Mono_C08_HLA-DQA1 and Mono_C09_C1QB as intermediate monocytes (CD14[+], FCGR3A[+]). Both Mono_C08_HLA-DQA1 and Mono_C09_C1QB highly expressed HLA-II class genes.

Mono_C08_HLA-DQA1 was specifically enriched in the IM group, with no upregulation observed in HLH compared to HV (Fig. 6d). Functional enrichment analysis of this cluster revealed that its upregulated genes were primarily associated with MHC class II antigen presentation pathways, including "antigen processing and presentation of exogenous peptide antigens" and "MHC class II protein complex assembly" (Fig. 6e). In contrast, downregulated genes were associated with humoral immune responses, microbial defense, and positive regulation of cytokine and inflammatory signaling (Fig. 6e). These findings suggest that monocytes in IM adopt an antigen-presenting phenotype, supporting MHC class II−mediated T cell activation while limiting inflammatory responses−potentially contributing to the self-limiting course of IM. In contrast, monocytes in HLH display a pro-inflammatory profile that may promote cytokine storm development.

Compared to HV (Fig. 7a), the upregulated DEGs in HLH participated in cell activation, positive regulation of response to external stimuli, positive regulation of cytokine production, and positive regulation of canonical NF-κB signal transduction. Compared to IM (Fig. 7b), the upregulated DEGs in HLH were enriched in pathways, such as inflammatory response, cellular response to cytokine stimulus, antigen processing and presentation, regulation of dendritic cell differentiation, and regulation of the viral life cycle.

Further analysis identified genes specifically upregulated or downregulated in the HLH group, revealing differences compared to HV and IM groups. IDO1 was upregulated in HLH, participating in pathways including negative regulation of leukocyte activation, negative regulation of cell-cell adhesion, inflammatory response, and positive regulation of cytokine production (Fig. 7c). This indicates its role in the pathogenesis of HLH. Additionally, genes involved in the positive regulation of the MAPK cascade and canonical NF-κB signal

transduction were also upregulated in HLH. In contrast, genes involved in the negative regulation of the MAPK cascade (*DUSP1, PTPRC, RGS2, TLR4*) were downregulated in HLH (Fig. 7d).

Enrichment analysis results showed that monocytes in HLH exhibited abnormally activated NF-κB signaling, MAPK signaling, and inflammatory responses. Consistent with transcriptomic data, phosphorylated p65 levels were elevated in HLH monocytes compared to IM, indicating activation of the canonical NF-κB pathway (Fig. 7e). Furthermore, Ucell scoring indicated that Mono_C06_CD14_IDO1 had the highest inflammation score (Fig. 7f), and this subgroup was enriched in the HLH group (Fig. 6d and Supplementary Fig. 6d). Moreover, across all monocyte subpopulations, inflammation scores in HLH were higher than in HV and IM groups (Fig. 7g).

Collectively, these findings indicate that monocytes may function as effector cells and contribute significantly to the inflammatory cytokine storm observed in HLH.

## IDO1[+] monocyte with specific activation revealed by trajectory analysis

To investigate the transitions between monocyte subpopulations, we utilized Monocle2 for pseudotime trajectory analysis[38]. This approach allowed us to arrange monocytes along a trajectory based on their expression profiles and transition dynamics. Monocytes were categorized into five cell groups, each representing a different state of differentiation, based on their gene expression patterns. According to the primary branching points, monocytes differentiated into two distinct branches, termed cell fate 1 and cell fate 2 (Fig. 8a and Supplementary Fig. 7a). The tissue distribution of these two fates differed significantly (Fig. 8b and Supplementary Fig. 7b, c), with cell fate 1 predominantly found in HLH, and Mono_C06_CD14_IDO1 cells clustering at the end of fate 1, indicating disease specificity of this differentiation path.

Subsequent analysis focused on gene expression patterns and pathways associated with changes along fate 1 (Fig. 8c). We observed that certain genes, including *IDO1* and *IL18*, were gradually upregulated along fate 1. These genes were associated with inflammatory responses and the positive regulation of cytokine production. In contrast, some genes that were downregulated along fate 1 were related to endocytosis (*ITGB1*) and the negative regulation of the MAPK cascade (*DUSP1, DUSP7*). Additionally, genes responding to interferon (*IFI27, IRF7, IFITM1, ISG15, IRF1, GBP1, IFNGR2*) were upregulated along fate 1, suggesting that cells differentiating along fate 1 are influenced by interferon stimulation, exhibiting pro-inflammatory characteristics with reduced phagocytic ability.

We then predicted transcription regulators for cells in fate 1, identifying the top 10 transcription factors with the highest activity scores: *MAFG, GTF2B, SOX6, TCF7L1, ETV7, IRF7, ZNF713, ZNF846, KLF5, and ZNF32* (Fig. 8d and Supplementary Fig. 7d). We defined the cell subgroup prior to the branch as Mono_S01, the subgroup on branch 2 as Mono_S02, and the subgroup on branch 1 as Mono_S03 (Fig. 8e). Comparing with the inflammation score plot (Fig. 8f), cells with high inflammation scores were primarily concentrated in the Mono_S03 subgroup, which was enriched in the HLH group (Fig. 8g and Supplementary Fig. 7e). The bubble chart illustrated the characteristic genes of Mono_S03. Notably, *IDO1* was highly expressed in Mono_S03, while

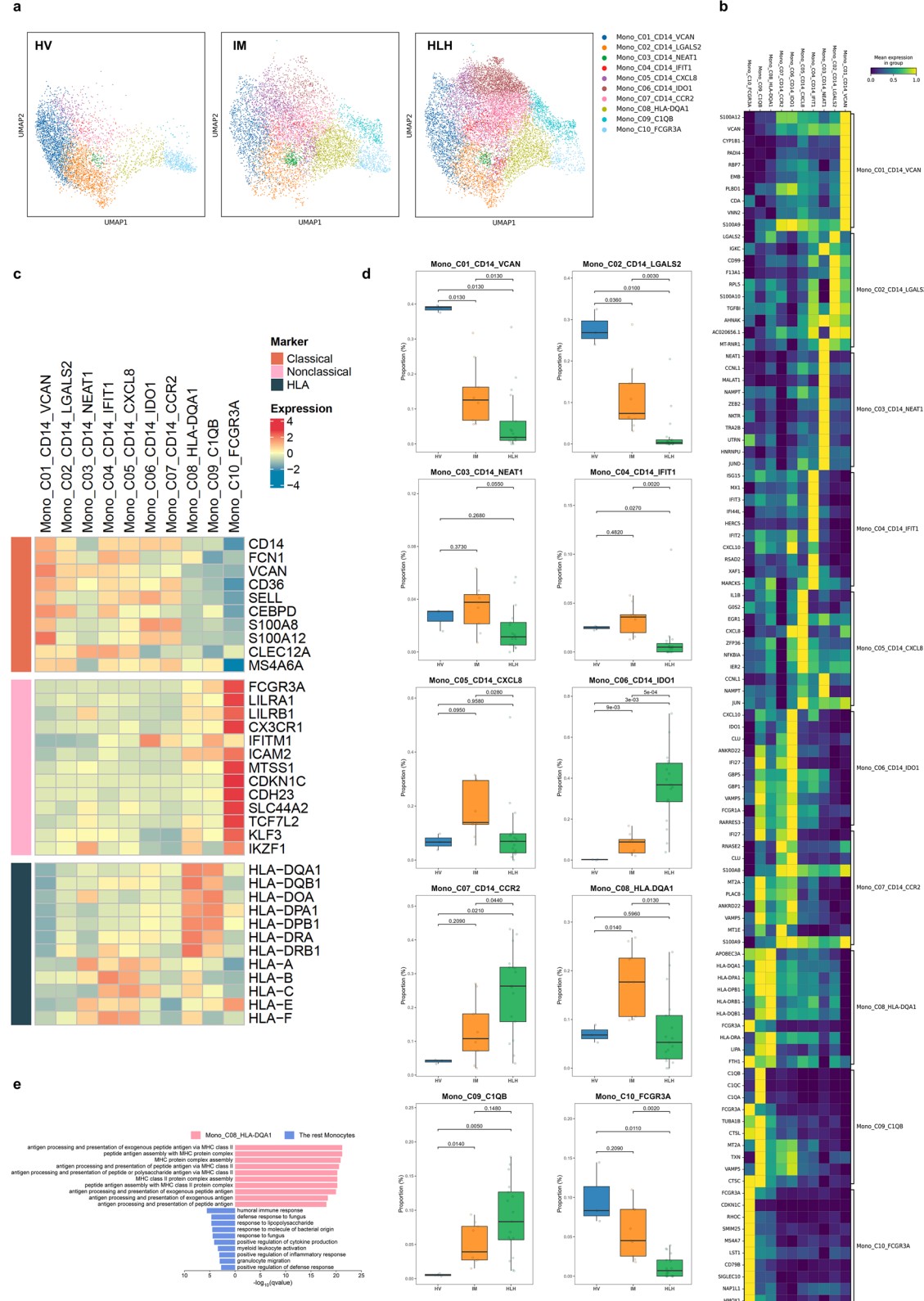

**Fig. 6 | Sub-clustering analysis on Monocytes. a** UMAP visualization of 5228 single cells from HV (*n* = 3), 7298 single cells from IM (*n* = 9), and 12,387 single cells from HLH (*n* = 17). The cells were divided into 10 clusters and color-coded accordingly. **b**, **c** Heatmap representing the average expression values of the highly expressed 10 genes (**b**) and functional genes (**c**) in each monocytes cluster. **d** Box plot depicting the percentage of each cell cluster in HV (*n* = 3), IM (*n* = 9), and HLH (*n* = 17) group,

with the median indicated by a horizontal line. Two-sided Wilcoxon tests were performed between groups and *P*-values were adjusted using the BH correction. **e** Enriched GO terms of differentially expressed genes between Mono_C06_HLA-DQA1 cluster and the rest monocytes. The enrichment results were produced by clusterProfiler and BH-corrected.

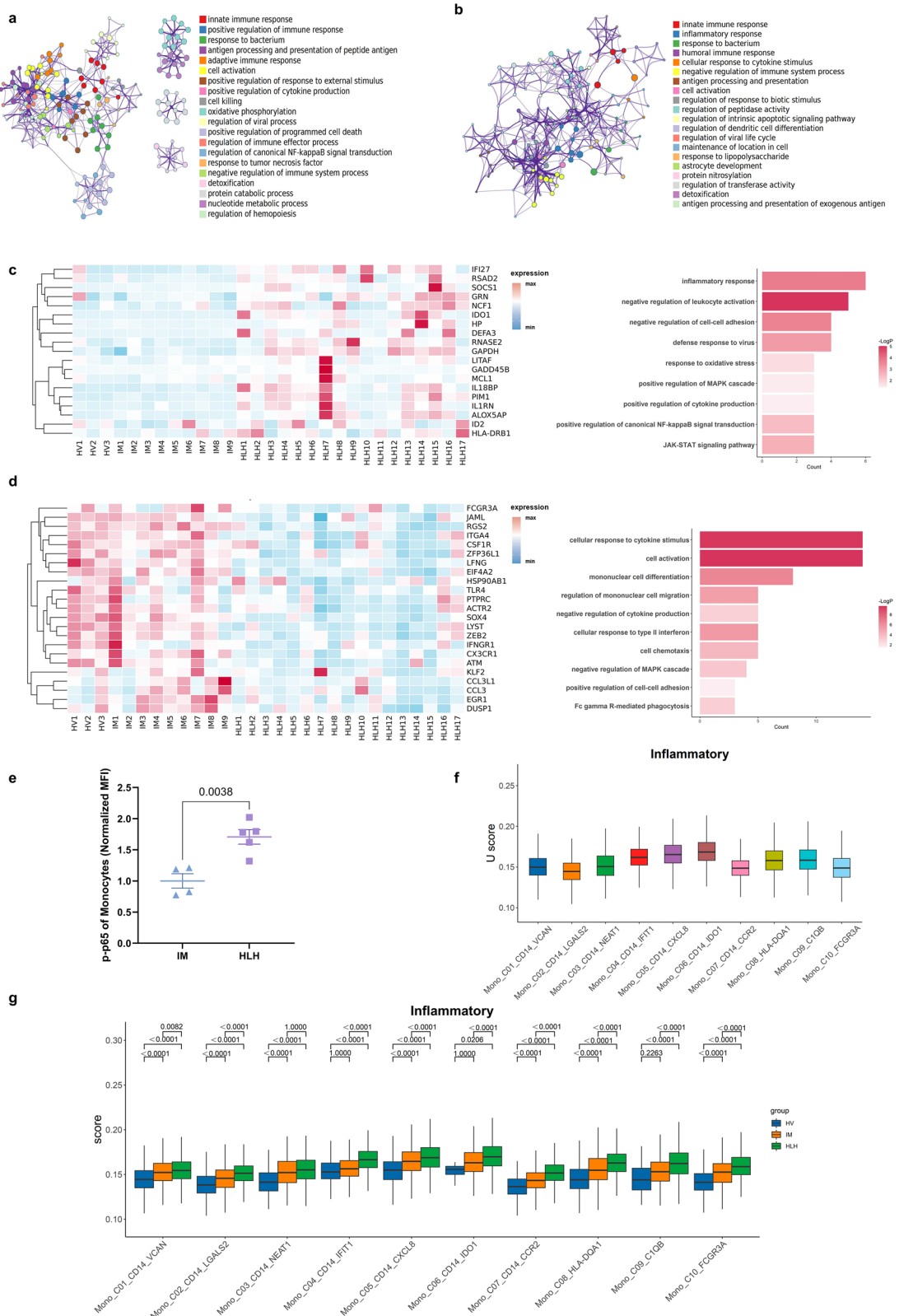

**Fig. 7 | Immunological characterization of Monocytes clusters.** Enriched GO terms for upregulated genes in HLH patients compared to HV (**a**) and IM (**b**). **c, d** The expression levels of specific genes in all samples and their enrichment in the GO and KEGG pathways. The enrichment results were produced by Metascape and BH-corrected. Compared to the HV group and IM group, these genes were upregulated (**c**) or downregulated (**d**) in the HLH group. In contrast, expression levels between the HV and IM groups either showed no significant differences or

exhibited a reverse trend. **e** Quantification of phosphorylated p65 levels (normalized MFI values) in monocytes from pediatric IM (*n* = 4) and HLH (*n* = 5) biologically independent samples. Two-sided Independent-samples *t*-test was applied. **f, g** The box plot shows the scoring of the inflammatory response set for each cell cluster among the three groups (HLH, HV, and IM). Comparisons were made using the two-sided Wilcoxon tests and *P*-values were adjusted using the BH correction.

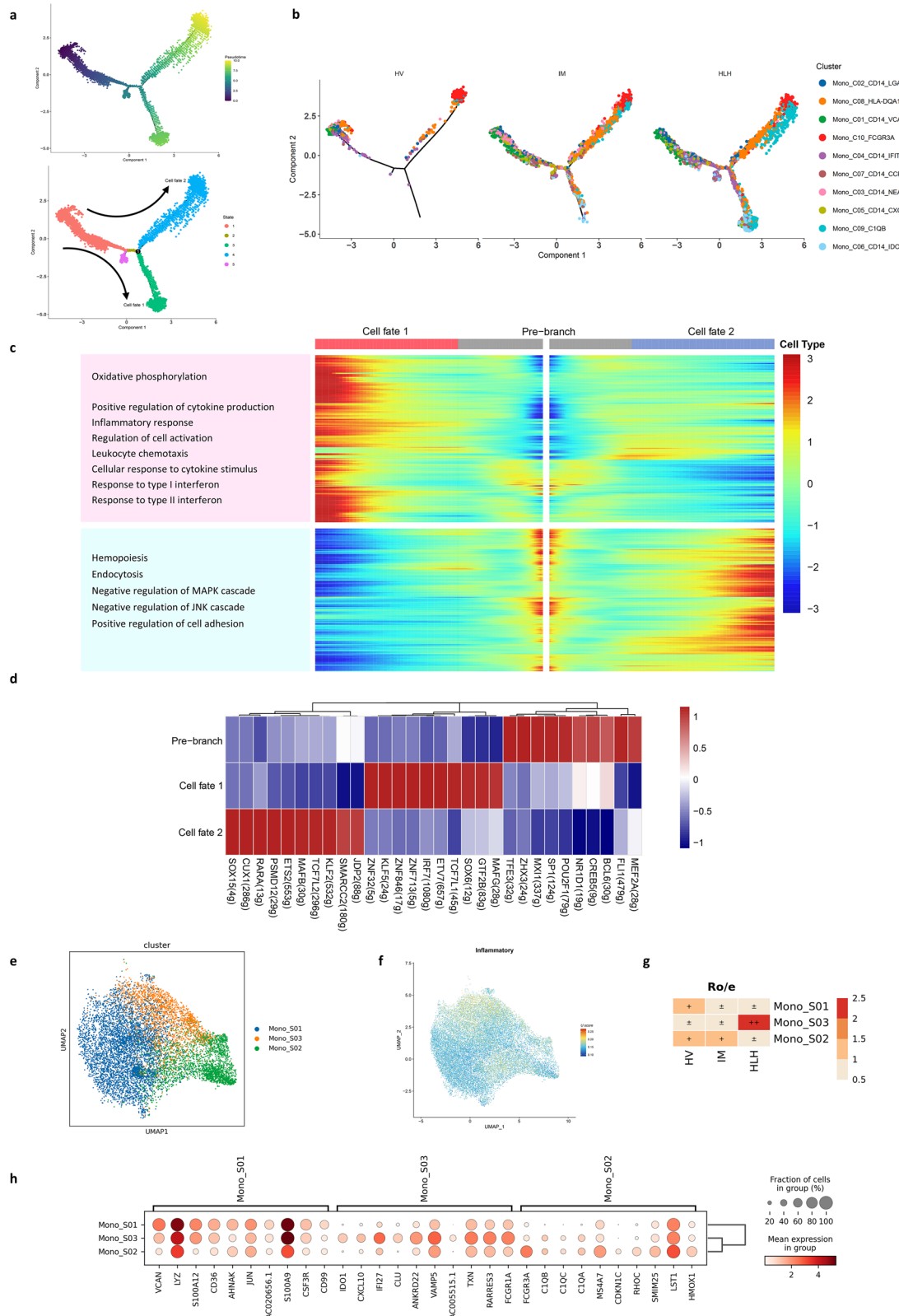

it was almost not expressed in the other two groups, making it a distinct marker for the Mono_S03 cluster (Fig. 8h).

### Validation of IDO1⁺ monocytes and L-kynurenine as new biomarkers for HLH

To verify the potential of IDO1⁺ monocyte populations as biomarkers for HLH, we conducted flow cytometry analysis on fresh peripheral blood samples collected from a validation cohort. This cohort included 6 healthy controls (HV), 4 pediatric patients with infectious mononucleosis (IM), 7 newly diagnosed EBV-HLH pediatric patients (HLH), and 3 paired EBV-HLH pediatric patients who achieved complete remission (CR) after standard treatment (HLH-T). The samples of HLH-T group were collected within 4–6 weeks after standard treatment. The remaining samples were collected within 0–3 days after admission

**Fig. 8 | Trajectory inference of monocytes differentiation. a** The trajectories of monocytes were visualized and colored according to pseudotime (left) and cell state (right). **b** Pseudotime trajectory analysis of monocytes among each group, where each dot signified an individual cell and was colored according to its cluster label. **c** Heatmap representation of GO analysis findings for signaling pathways unique to fate 1, plotted along pseudotime. **d** Heatmap displaying the top 10 regulons in each cell type, with the average AUC (Area Under the Curve) matrix clustered. Numbers in parentheses indicate the number of target genes for each transcription factor. This visualization reveals the transcriptional regulators most strongly associated with each cell type, offering clues to the transcriptional control mechanisms at play during monocyte differentiation. **e** UMAP visualization showing monocyte

subpopulations clustered by fate. Mono_S01 represented cells on the pre-branch, indicating an undifferentiated or initial state. Mono_S02 represents cells on the fate 2 branch, and Mono_S03 represents cells on the fate 1 branch, indicating differentiated states along two distinct developmental paths. The presence of cells on the fate 1 and fate 2 branches were denoted by Mono_S03 and Mono_S02, respectively, demonstrating different stages of differentiation across two unique developmental trajectories. **f** UMAP visualization displaying the scoring of the inflammatory response set for each monocyte. **g** Quantification of cell subpopulation enrichment based on the Ro/e value for each cell population. **h** Dot plot showing the highly variable genes for three cell classifications. The diameter of each circle reflected the percentage of cells expressing the gene in the subtype, and the color denotes the mean expression.

(acute phase, prior to treatment). The flow cytometry analysis revealed that traditional monocyte markers, CD14 and CD16, were unable to differentiate HLH patients from HVs and IM patients. However, there was a higher proportion of IDO1⁺ monocytes in HLH patients compared to the other groups. There was no significant difference in the proportion of IDO1⁺ monocytes among HV, IM, and HLH-T groups (Fig. 9a). This suggests that the level of IDO1⁺ monocytes increases during the acute phase of HLH and decreases to levels comparable to healthy children after treatment. The analysis of paired samples also supported this conclusion (Fig. 9b). Therefore, IDO1⁺ monocytes can serve as biomarkers for the diagnosis and treatment of HLH.

To assess whether IDO1 expression reflects general monocyte activation or represents a distinct feature of EBV-HLH, we analyzed its association with established activation markers (HLA-DR, CD163, and CX3CR1) using flow cytometry in a new validation cohort comprising IM and EBV-HLH samples. A significantly higher proportion of IDO1⁺ monocytes was observed in the HLH group compared to the IM group. In contrast, HLA-DR expression was uniformly high in both groups, with no significant differences in the proportions of HLA-DR⁺, CD163⁺, or CX3CR1⁺ monocytes (Fig. 9c). Further analysis within the HLH group revealed no significant differences in HLA-DR⁺, CD163⁺, or CX3CR1⁺ populations between IDO1⁺ and IDO1⁻ monocytes. Similarly, the frequency of IDO1⁺ cells did not vary across subgroups defined by the expression of these markers (Fig. 9d). These results suggest that IDO1 upregulation in HLH occurs independently of conventional monocyte activation states. The flow gating strategy and FACS plots for staining were in Supplementary Fig. 8.

IDO1 is the key rate-limiting enzyme that metabolizes tryptophan into kynurenic acid[39]. To further investigate this, we collected serum samples for targeted tryptophan metabolomics sequencing. Consistent with the results from single-cell sequencing and flow cytometry, the concentration levels of L-kynurenine, the direct metabolite of IDO1, were specifically elevated in the HLH group and reduced in the HLH-T group (Fig. 9e). We further investigated in vitro whether kynurenine (KYN) affects the inflammatory responses of monocytes and cytotoxic T lymphocytes (CTLs). We conducted endogenous enhancement (via IDO1 overexpression) and exogenous kynurenine (KYN) supplementation treatments in THP1 and U937. qRT-PCR analysis confirmed that KYN significantly upregulated pro-inflammatory cytokine expression in these monocytes (Fig. 9f). Similarly, exogenous KYN stimulation enhanced pro-inflammatory responses in primary T cells and NK92 cells (Fig. 9g).

### Recognition of cellular communication in monocytes

To gain deeper insight into the cellular communication between monocytes and other immune cells, we utilized the CellphoneDB algorithm to investigate interactions among cell populations within the HV and HLH groups. This analysis revealed that interactions between monocyte populations and DCs, NK cells, NKT cells, and CD8⁺ T cells were particularly abundant, with the highest number of interactions occurring between monocytes and DCs (Supplementary Fig. 9a).

Significant changes were observed in the interactions between cells in the HV and HLH groups. Compared to HV, the ligand-receptor

interactions in monocytes were increased in HLH patients. In HLH, monocytes highly expressed receptors, such as *CCR1, TNFRSF1A, TNFRSF1B*, and *TNFRSF14*, which can receive stimulation from various cytokines secreted by other cell types. Additionally, monocytes in HLH highly expressed cytokine ligands, such as *CXCL10, CXCL8, CCL3, TNF*, and *TNFSF10*, which can stimulate corresponding receptors on other cell types (Fig. 10a–c and Supplementary Fig. 9b, c).

Overall, these results indicate the enhanced communication between monocytes and other immune cells during EBV-HLH progression.

## Discussion

EBV-HLH involves a highly complex interplay of genetic, immunological, and viral factors. The disease is characterized by an excessive and dysregulated immune response, leading to a cytokine storm that can cause severe tissue damage[7]. In this study, we used single-cell RNA sequencing to analyze the immune characteristics in pediatric EBV-HLH patients and controls (IM patients and healthy children). Our findings provided new insights into the pathogenesis and biomarkers of EBV-HLH.

At the transcriptional level in T cells and NK cells of EBV-HLH patients, there was no observed downregulation of cytotoxic genes, such as *PRF1* and *GZMB*. This finding suggests that the decreased CTL killing ability in EBV-HLH is not due to a deficiency in cytotoxic-related functions, which is a notable difference from familial HLH. We further revealed that the *ANXA1* gene, which plays a critical role in inflammatory response[40], was significantly downregulated in HLH. The downregulation of *ANXA1* could alter T cell activation signals, thereby affecting T cell receptor-mediated signal transduction and ultimately leading to an excessive inflammatory response and increased risk of tissue damage[41]. The downregulation of genes involved in immune surveillance and killing functions, such as *RASGRP1, CD81*, and *CD160*, impairs the ability of NK cells to clear viruses in HLH. Moreover, the abnormal downregulation of genes involved in B cell receptor signaling and cell differentiation in B cells may contribute to the reduction of plasma cells, indicating an attenuation of humoral immunity in the HLH group. Targeted restoration of their function could theoretically enhance immune surveillance against EBV, which may represent a potential therapeutic strategy.

At the signal transduction level, we found that abnormal activation of the NF-κB and MAPK pathways in T cells, B cells, NK cells, and monocytes. These pathways are crucial for inflammation and cytokine production[42,43], and their dysregulation likely contributes to the excessive inflammatory response observed in hemophagocytic syndrome. Although EBV can influence disease progression through NF-κB pathway activation[3], current studies have not focused extensively on the roles of the NF-κB and MAPK pathways in HLH. Our results highlight the aberrant activation of these pathways in EBV-HLH, suggesting that this activation is a key factor in the excessive inflammatory response. Whether targeting the abnormal activation of the NF-κB and MAPK pathways can treat secondary HLH warrants further clinical study.

Our findings highlight that monocytes are the primary effector cells in HLH's cytokine storm. Among all immune cells, monocytes had the highest inflammatory scores, and within all monocyte subgroups,

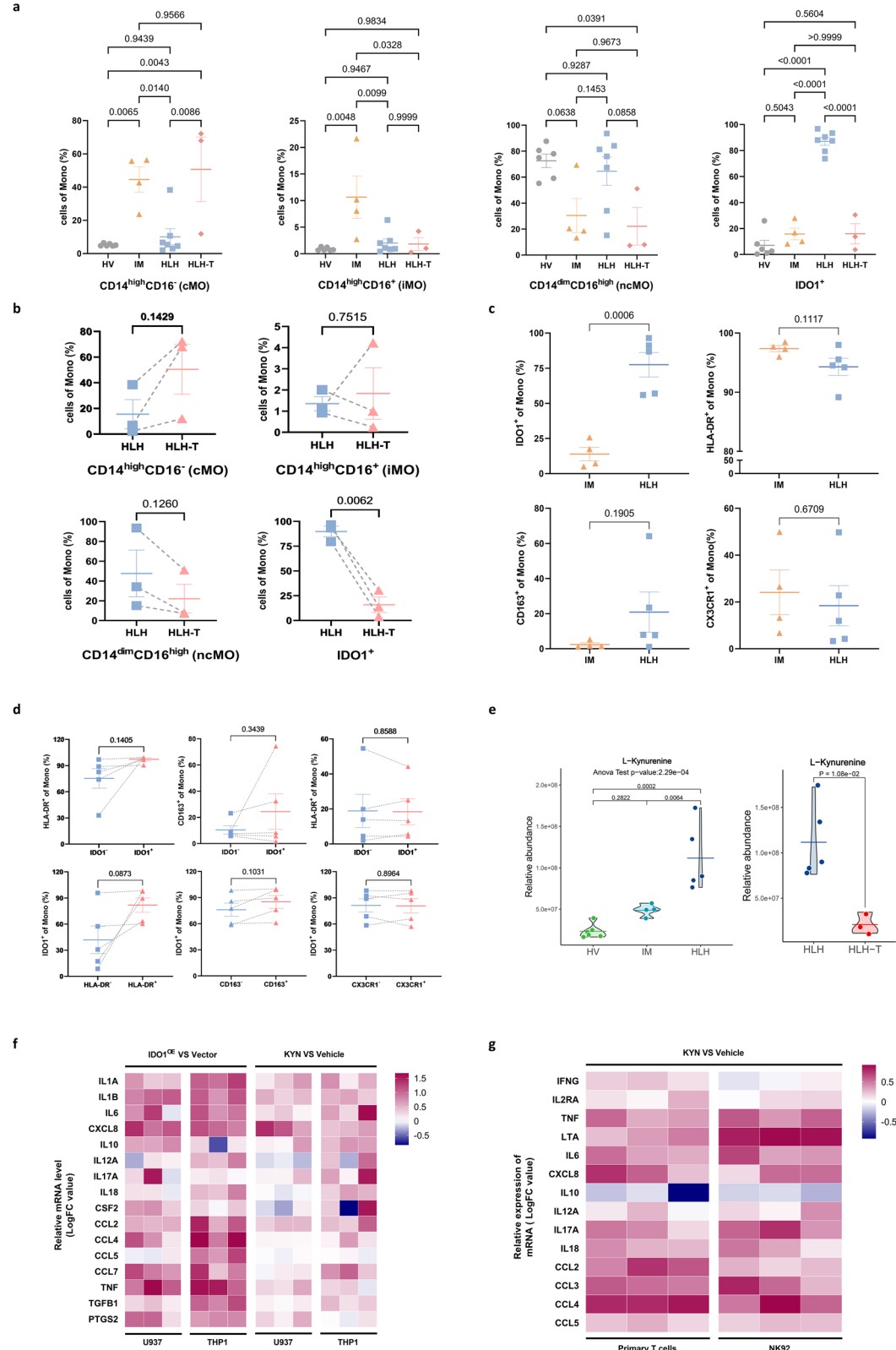

the inflammatory score in HLH was higher than that in the HV and IM groups. Trajectory analysis identified a novel activated subgroup of IDO1⁺ monocytes, characterized by high inflammatory scores and a strong interferon response. IDO1, the principal rate-limiting enzyme catalyzing tryptophan (TRP) metabolism along the kynurenine pathway outside the liver[44-46], has been extensively studied in the context of the tumor microenvironment. Tumors can create an

immunosuppressive environment by overexpressing IDO1, inhibiting anti-tumor immune responses. The mechanisms involved include inducing inactivation of T and NK cells, as well as activation and expansion of Treg cells[47]. Thus, IDO1 is considered a tumor immune checkpoint, and the efficacy of IDO1 inhibitors, alone or in combination with other drugs (e.g., immune checkpoint inhibitors), has been extensively studied in cancer immunotherapy[48,49].

**Fig. 9 | Validation analysis of IDO1+ monocytes as new biomarkers for HLH.**
**a, b** The proportions of IDO1+ monocytes in the validation biologically independent samples (n = 6 in HV, n = 4 in IM, n = 7 in HLH, n = 3 in HLH-T) by flow cytometry. The discrepancies between multiple groups were assessed using one-way ANOVA along with two-sided Tukey–Kramer post hoc testing (**a**). Two-sided Paired t-test was used for the three paired samples of HLH and HLH-T (**b**). **c** The proportions of IDO1+, HLA-DR+, CD163+, and CX3CR1+ monocytes in the IM (n = 4) and HLH (n = 5) biologically independent samples by flow cytometry. Two-sided Independent-samples t-test was applied. **d** In monocyte subsets of HLH patients (n = 5), comparative analysis of HLA-DR+/CD163+/CX3CR1+ proportions between IDO1+ and IDO1− subpopulations (upper panel) and evaluation of IDO1+ proportions across positive/negative of HLA-DR, CD163, and CX3CR1 subgroups (lower panel). Two-sided Paired t-test was applied. **e** Box plot depicting the relative expression levels of L-Kynurenine across biologically independent sample groups (n = 6 in HV, n = 4 in IM, n = 5 in HLH). One-way ANOVA along with two-sided Tukey–Kramer post hoc testing was employed to assess the significance between multiple groups. **f** In THP1 and U937 cell lines, mRNA levels of inflammation-related cytokines were quantified via qRT-PCR following either IDO1 overexpression or exogenous KYN treatment for 48 h. **g** In primary CD3+ T cells and NK92 cell line, mRNA levels of inflammation-associated cytokines were quantified by qRT-PCR following exogenous KYN treatment for 48 h. The data were presented as means ± SEM (error bar) in (**a–d**).

Beyond tumors, increasing evidence suggests that IDO1 also plays an important role in viral infections, including HIV[50,51], influenza[52], hepatitis B[53], and C[54]. In the context of pathogen infection, IDO1 activity is pleiotropic: it can directly inhibit the replication and spread of certain pathogens, while also acting on host cells to suppress immune responses, thereby promoting viral activities[45,55]. For example, HIV-1 viruses exploit the immunosuppressive activity of IDO1 to establish chronic HIV infection. The dual role of IDO1 in infectious diseases might be associated with the type of pathogen and the stage of disease development, indicating the complexity of its mechanisms of action. However, there is currently limited research on the role of IDO1 in EBV infectious diseases. A recent study of EBV-associated posttransplant lymphoproliferative disorders (PTLD) identified IDO1 as a key target for the transformation of EBV-infected B cells[56]. It has been confirmed in animal models that the use of IDO1 specific inhibitor in the early stages reduced viremia and lymphoma[56]. This result suggests the strong potential of IDO1 as an immune metabolic intervention target in EBV-associated diseases.

We demonstrated the involvement of IDO1 in EBV-HLH and suggest that targeting IDO1+ monocytes could be a potential therapeutic strategy for managing the excessive inflammatory response in HLH. IDO1+ monocytes and L-kynurenine specifically increased in the EBV-HLH group and decrease in patients who have achieved CR after treatment. In vitro studies using monocytic and cytotoxic T cell lines demonstrated that L-kynurenine promotes the production of multiple pro-inflammatory cytokines. These findings suggest that L-kynurenine may contribute to the amplification of inflammatory responses, providing new insight into its potential role in the pathogenesis of HLH.

In summary, we used single-cell RNA sequencing to generate detailed immune profiles of children with EBV-IM, EBV-HLH, and asymptomatic EBV carriers at single-cell resolution. Elevated IDO1 expression and kynurenine levels may serve as potential auxiliary biomarkers for distinguishing EBV-HLH. However, the biomarker validity requires further confirmation through prospective multicenter cohorts and functional experiments. In conclusion, these findings enhance our understanding of the immune response mechanisms following EBV infection and contribute to identifying new treatment strategies for EBV-HLH.

## Methods
### Patient cohorts and study design
This study was approved by the Institutional Review Boards of the third Xiangya Hospital of Central South University. Informed consent was obtained from parents/guardians. This study performed scRNA-seq analysis on the PBMCs of 29 children, including 3 healthy volunteers (HV group), 9 patients with infectious mononucleosis (IM group), and 17 patients with EBV-HLH (HLH group). Their baseline characteristics were summarized in Supplementary Table 1. Detailed clinical information could be obtained from Supplementary Table 2. All subjects were tested for EBV DNA before enrollment to confirm EBV infection as positive, and EBV-HLH patients underwent whole-exome sequencing before inclusion to confirm the absence of pathogenic gene mutations associated with primary HLH and immunodeficiency. The relevant gene panel was provided in Supplementary Table 3.

### PBMC processing
The PBMCs were isolated using Ficoll-Paque Plus medium (Cat#7144002, GE Healthcare, USA) via density gradient centrifugation. Subsequently, they were washed with PBS. Red Blood Cell Lysis Buffer (RCLB, Singleron Biotechnologies Co., Ltd, Nanjing, China)[14] was used to remove red blood cell. Following red blood cell removal, the sample underwent a further centrifugation at 4 °C and 400 g for 10 min to isolate the PBMCs. PBMCs were resuspended in PBS. Ultimately, cell viability was assessed through Trypan Blue staining and microscopic examination.

### RT & amplification & library construction
The concentration was $2 \times 10^5$ cells/mL of single-cell suspension, and it was introduced into the microfluidic chip using the Singleron Matrix single-cell processing system[57]. Subsequently, barcode beads were extracted from the chip, resulting in reverse transcription and PCR amplification of the separated mRNA. The single-cell RNA sequencing libraries were generated with the guidelines provided by Singleron, and individual libraries were sequenced using the Illumina NovaSeq 6000 platform.

### Raw read data analysis
The CeleScope pipeline (version 1.9.0, accessible at https://github.com/singleron-RD/CeleScope) was employed to process raw reads, resulting in the production of gene expression matrices. Initially, CeleScope was utilized to filter out low-quality reads. This step was followed by the extraction of cell barcodes and Unique Molecular Identifiers (UMIs). Subsequently, reads were mapped to the GRCm38 reference genome using STAR software (version 2.6.1a)[58], which includes annotation information from Ensemble version 92. Then, featureCounts software (version 2.0.14)[59] was used to determine UMI and gene counts for each cell.

### Single-cell gene expression quantification and determination of cell types
We used the "DoubletFinder" R package (available at https://github.com/chris-mcginnis-ucsf/DoubletFinder) to address the issue of erroneous doublets formed during cell encapsulation in droplets. Certain clusters are essentially doublets comprising multiple cell types, and no single cell type has been observed to strongly express these markers simultaneously. We systematically eliminated doublets from each sample, setting the anticipated doublet rate at 5% and adhering to the package's default settings for other parameters. Post filtration, the cells that met the criteria were deemed to be individual single cells. The gene expression matrices obtained from filtered cells were merged and subjected to logarithmic normalization and linear regression using Seurat software.

We mitigated batch effects using Canonical Correlation Analysis (CCA) and the RunUMAP function within Seurat. Using the FindClusters function in Seurat, cell clusters were identified, and then classified

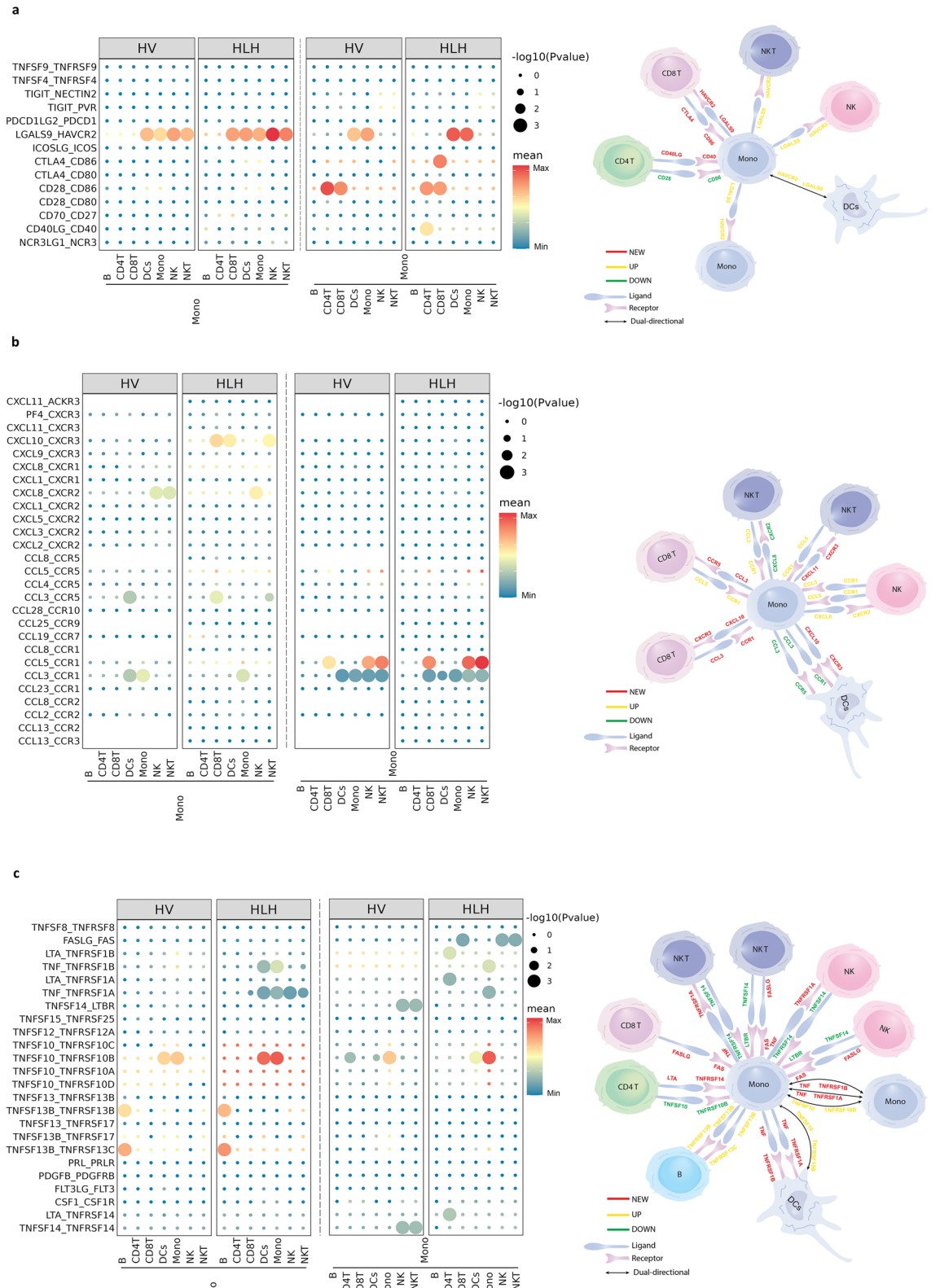

**Fig. 10 | Intercellular interaction alterations among cell types between HV and HLH sample groups.** Predicted cellular interactions between monocytes and other cell types, represented by bubble charts (left) and corresponding schematic diagrams (right), including immune checkpoint interaction pairs (**a**), chemokine interaction pairs (**b**), and cytokine interaction pairs (**c**). The circle size in this illustration reflected the significance *P*-value of the ligand-receptor axis, and the color indicated the specificity of the interactions. The interaction results were produced by CellPhoneDB.

into different major cell types based on the established marker gene, including CD4 T cells (*CD3D, CD3E, CD4*), CD8 T cells (*CD3D, CD3E, CD8A, CD8B*), γδ T cells (GDT; *TRDC, TRGC1*), natural killer (NK) cells (*KLRF1*), NKT cells (*CD3D, CD3E, KLRF1*), B cells (*CD79A, CD79B, MS4A1*), plasma B cells (*MZB1, XBP1*), monocytes (*LYZ, CD14*), cDC (*FCER1A, CD1C*), and pDC (*LILRA4*).

By analyzing the average expression of specific gene sets within each major cell type, they were classified into various specific cell subtypes. Seurat's FindAllMarkers function was used to compare the expression profiles of each subcluster with those of other subclusters to determine the marker genes for each subcluster. Differential expression analysis employed a two-sided non-parametric Wilcoxon rank-sum test. If the *P*-value indicated significance after Bonferroni adjustment, with a threshold below 0.05, it was deemed to have significant differences.

If a cell cluster was found to have multiple well-defined marker genes, it was regarded as contaminated and thus excluded from further analysis. The criteria for selecting marker genes included: (1) ranking prominently in the analysis of differential gene expression within specific cell clusters, (2) demonstrating strong specificity in gene expression, manifested by a high expression ratio within the target cluster compared to others, and (3) being supported by literature as either a confirmed marker gene or functionally associated with the cell type.

### Calculation of functional module scores
To assess the functional capabilities of cell clusters, we calculated the scores of functional modules for the cell cluster, using the "Ucell" R package (available at https://github.com/carmonalab/UCell) at single cell level. The functional modules including cytokine, inflammatory, naïve, cytotoxicity and exhausted scores. The relevant genes are detailed in the Supplementary Table 4.

### DEGs identification and functional enrichment
We utilized the FindMarkers function in Seurat for performing DEGs analysis, employing the default settings of the Wilcoxon test. The Benjamini-Hochberg (BH) correction method was applied to evaluate the false discovery rate (FDR). For the segregation of DEGs, criteria were set, including a minimum log2(fold change) of 0.5 and a maximum FDR of 0.01. Gene enrichment analysis, including Gene Ontology (GO) and KEGG Pathway, was conducted using the Metascape website (http://metascape.org/) and "clusterProfiler" R package[60] (available at https://bioconductor.org/packages/release/bioc/html/clusterProfiler.html)

### Developmental trajectory inference
We utilized Monocle2 for trajectory analysis of individual nucleus cells to determine lineage differentiation among various immune cells. Initially, the clustered data was transferred from Seurat to Monocle2 (version 2.8.0; http://cole-trapnell-lab.github.io/monocle-release/monocle2/). Subsequently, the dpFeature method in Monocle2 was employed in conjunction with the differentialGeneTest function to organize cells based on inter-cluster gene expression variability. Following this, utilizing default parameters, after dimensionality reduction and cell ordering, the trajectory of cell differentiation was inferred.

### Cellular communication analysis
CellPhoneDB (version 2.0.6; https://github.com/Teichlab/cellphonedb) were employed to conduct ligand-receptor analysis, investigating potential intercellular signaling between different cell types. Gene expression matrices from CD8 T, CD4 T, NK, NKT, B cells, monocytes, and DCs were used as inputs. We calculated the interaction strength between two cell types based on the average expression levels of ligands and receptors. Comparisons were made between the average values of specific receptor-ligand pairs and the average distribution of randomly permuted pairs to determine significance, where significance was indicated by *P* < 0.05.

### Cell culture
The complete medium to maintain the cell lines were as follows: Primary T cells: T Cell Expansion Medium (Cat#10981, STEMCELL Technologies, Canada) + 100 U/mL recombinant IL-2 (Cat#GMP-C013, Novoprotein, Shanghai, China) + 25 μL/mL Human CD3/CD28 T Cell Activator (Cat#10971, STEMCELL Technologies, Canada). NK-92: SuperCulture N500 Medium (Cat#6113031, Dakewe, Shenzhen, China) + 10% fetal bovine serum (Cat#BC-SE-FBS01C, BioChannel Biological Technology Co., Ltd, Nanjing, China) + 200 U/mL recombinant IL-2 + 1% penicillin/streptomycin (Cat#GA3502, Genview, China); U937 and THP1: RPMI-1640 medium (Cat#GR3101, Genview, China) + 10% fetal bovine serum + 1% penicillin/streptomycin.

### Quantitative reverse transcription-polymerase chain reaction (RT-qPCR)
The total cellular RNA was extracted using the RNeasy Kit (Cat#RNA-fast200, Fastagen, Shanghai, China) following the manufacturer's instructions. Then, 1000 ng of total RNA was reverse transcribed into complementary DNA (cDNA) using the Evo M-MLV RT Kit (Cat#AG11705, ACCURATE BIOTECHNOLOGY(HUNAN) CO., LTD, Changsha, China). Quantitative real-time RT-PCR was performed using the SYBR Green Pro Taq HS qPCR kit (Cat#AG11733, ACCURATE BIOTECHNOLOGY(HUNAN) CO., LTD, Changsha, China) and the Roche LightCycler 480. Primer information is available in Supplementary Table 5. The expression levels of genes were normalized to the expression of *ACTB*.

### Statistics analysis
All statistical analyses were conducted using GraphPad Prism or R. For boxplots, the horizontal line within each box represents the median, the top and bottom of each box indicate the 75th percentile (Q3) and 25th percentile (Q1), and the whiskers extend to the most extreme data points within $1.5 \times IQR$ from Q1 and Q3. For comparisons between two groups, independent-sample *t*-tests or paired *t*-tests were used for normally distributed variables, and Wilcoxon rank-sum tests were used for non-normally distributed variables. For comparisons among multiple groups, one-way analysis of variance (ANOVA) with Tukey–Kramer post hoc tests was performed. $P < 0.05$ were considered statistical significance; "ns" indicates not significant; $*P < 0.05$; $**P < 0.01$; $***P < 0.001$; $****P < 0.0001$.

### Reporting summary
Further information on research design is available in the Nature Portfolio Reporting Summary linked to this article.

## Data availability
The raw sequencing data have been deposited in the GSA-Human database under the HRA010943. All data are included in the Supplementary Information or available from the authors, as are unique reagents used in this Article. The raw numbers for charts and graphs are available in the Source Data file whenever possible. Source data are provided with this paper.

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

## Acknowledgements

This study was supported by the following grants: National Key Clinical Specialty Scientific Research Project, Z2023023 (M.H.Y.); National Natural Science Foundation of China, 82270185 (M.H.Y.); the Science and Technology Innovation Program of Hunan Province, 2022RC3077 (M.H.Y.); Health Research Project of Hunan Provincial Health Commission, W20242008 (M.H.Y.); National Natural Science Foundation of China, 82400213 (Z.L.); Joint Medical Research Project between Chongqing Science and Technology Bureau and Health Commission, 2020FYYX050 (J.W.X.); Medical Research Project of Chongqing Municipal Health Commission, 2022MSXM069 (J.W.X.) and China Primary Health Care Foundation, xyxthjb-2023-004 (L.L.). The funders had no role in the study design, data collection and analysis, decision to publish, or preparation of the manuscript. We would like to thank Shanghai BioProfile Biotechnology (Shanghai, China) and Singleron Biotechnologies Co., Ltd (Nanjing, China) for technical supports.

## Author contributions

Conceptualization: M.H.Y., D.W. (Dao Wang), W.Z., Y.C.Z.; Methodology and Resources: J.S., Y.Y.H., J.W.X., F.L., K.K.C., B.Y.G., Y.L.H., J.W., Y.C.W., R.H., D.W. (Dao Wang); Visualization: J.S., H.Z.; Funding acquisition: M.H.Y., J.W.X., L.L.; Project administration: M.H.Y., D.W. (Dao Wang); Writing–original draft: J.S.; Writing–review & editing: M.H.Y., D.L.T., H.Z., Z.L., D.W. (Dan Wang), L.P.L., Y.Y.H., S.F.W.

## Competing interests

The authors declare no competing interests.

## Additional information

[1]Department of Pediatrics, Third Xiangya Hospital, Central South University, Changsha, Hunan, China. [2]Hunan Clinical Research Center of Pediatric Cancer, Changsha, Hunan, China. [3]Furong Laboratory, Changsha, Hunan Province, China. [4]Department of Pediatrics, The First Affiliated Hospital of Guangxi Medical University, Nanning, Guangxi, China. [5]Department of Pediatrics, The Xiangya Hospital, Central South University, Changsha, Hunan, China. [6]Department of Pediatrics, Children Hospital of Chongqing Medical University, Chongqing, China. [7]Department of Hematology and Oncology, Children's Hospital Affiliated to Shandong University and Jinan Children's Hospital, Jinan, Shandong, China. [8]Department of Pediatric Hematology and Oncology, School of Medicine, Children's Medical Center of Hunan Provincial People's Hospital of the First-Affiliated Hospital, Changsha, Hunan, China. [9]Department of Pediatrics, The First Affiliated Hospital of Xiamen University, Xiamen, Fujian, China. [10]Health Management Center, Chengdu Women's and Children's Central Hospital, School of Medicine, University of Electronic Science and Technology of China, Chengdu, Sichuan, China. [11]Department of Pharmacy, Third Xiangya Hospital, Central South University, Changsha, Hunan, China. [12]Cancer Research Institute, School of Basic Medical Sciences, Central South University, Changsha, Hunan, China. [13]State Key Laboratory of Experimental Hematology, National Clinical Research Center for Blood Diseases, Haihe Laboratory of Cell Ecosystem, Institute of Hematology and Blood Diseases Hospital, Chinese Academy of Medical Sciences and Peking Union Medical College, Tianjin, China. [14]Department of

Pediatrics, Xiangxi Autonomous Prefecture People's Hospital, Jishou, Hunan, China. [15]Department of Pediatrics, Liuyang people's hospital, Changsha, Hunan, China. [16]Department of Pediatrics, The First People's Hospital of Changde City, Changde, China. [17]Department of Surgery, UT Southwestern Medical Center, Dallas, TX, USA. [18]Department of Pediatrics, The First Affiliated Hospital of Zhengzhou University, Zhengzhou, Henan, China. [19]These authors contributed equally: Jie Shen, Yunyan He. ✉e-mail: deai315@163.com; yangminghua@csu.edu.cn

