## [Transparent Peer Review file · Nature Communications]

Biomarkers of Pediatric Epstein-Barr Virus-Associated Hemophagocytic Lymphohistiocytosis through Single-Cell Transcriptomics

Corresponding Author: Professor Minghua Yang

Version 0:

Reviewer comments:

Reviewer #1

(Remarks to the Author)

Shen et al. performed single-cell RNA sequencing to reveal the immune landscape in pediatric EBV-HLH. They found excessive cytokine secretion by T and NK cells along with a shift in monocyte differentiation towards an inflammatory phenotype. Moreover, L-kynurenine, regulated by IDO1, was specifically elevated in EBV-HLH, highlighting their potential as therapeutic targets. Overall, this manuscript is well-written and provides new insights into the pathogenesis of EBV-HLH. Specific comments are as follows.

1. EBV-HLH is related to primary EBV infection in most pediatric cases. Therefore, it is important to distinguish between primary EBV infection and reactivation. Serological data on EBV should be provided. Serological data is also important to discuss a potential weakening of humoral immunity in patients with HLH as stated in the liens 138–139. Furthermore, patients with chronic active EBV infection (CAEBV) can develop EBV-HLH. Were CAEBV cases included in this study? If so, the pathogenesis of EBV-HLH related to CAEBV would be different from other EBV-HLH.

2. In Figure S2C, the distribution of the T-cell phenotype of HLH5 is similar to that of IM (high proportion of CD8_STMN1). This patient might be classified as IM. Are there any relationships between the distribution of T-cells or other immune cell phenotypes and clinical course?

3. The effect of L-kynurenine on NK cells is shown in Figure 6. Authors have concluded that L-kynurenine in HLH may downregulate NK cell activation and the production of pro-inflammatory cytokines. On the other hand, pro-inflammatory cytokines can be produced by different immune cell types such as monocytes lymphocytes in EBV-HLH. Does L-kynurenine have the same effect on other immune cells?

4. The progression of EBV-HLH can differ, and two fatal cases were included in this study. Were there any distinct immunological features in these cases?

5. Single-cell sequencing analysis was performed using a small number of PBMC samples of patients with EBV-HLH in previous reports (Liu et al. PMID: 31914172 and Suzuki et al. 38642164). Are there any similarities or differences in immunological features compared to the previous reports?

6. The raw sequencing data should be accessible to the public.

Reviewer #2

(Remarks to the Author)

In this study, the authors performed single cell RNA sequencing to compare the transcriptomics of children with IM and EBV-HLH to identify unique immunological characteristics of pediatric EBV-HLH. Patients with EBV-HLH exhibited increased CD8+ T cells and decreased plasma cells, with excessive cytokine secretion by T and NK cells. Monocyte differentiation towards an inflammatory phenotype, and the aggregation of IDO1+ monocytes were also identified. They also showed that

L-kynurenine is selectively increased in sera from EBV-HLH patients and propose IDO and L-kynurenine as possible therapeutic targets. Overall, the analysis is well done and the study succeeded in providing immunological landscape of EBV-HLH at single cell level.

However, admitting advantages of the current study in providing detailed information on minor cell populations which are not readily identified by the conventional analyses, the analysis is superficial and lacks depth. Activation of CD8+ T cells and increased inflammatory cytokine production by T and NK cells in EBV-HLH has been well described. It has also been reported that activation of RIG-I pathway in monocyte-derived macrophages by viral EBER1 delivered from exosomes derived from EBV-infected cells induces IDO activation. There are many reports regarding the role of IDO in pathogenesis of EBV-associated diseases. Therefore, this paper fails to provide pivotal findings in immunological pathology of EBV-HLH.

Major concerns:

1. Diagnostic criteria of IM and EBV-HLH for this study is not provided. EBV-infected cell populations in each EBV-HLH patient should be provided.
2. Detailed immunophenotypic analysis of each patient depending on the disease severity and infected cell types should be performed.
3. It is more interesting and important to know the differences between specific cell populations infected or not-infected by EBV. In other words, what differences can be found in EBV-infected and EBV-uninfected CD8+, CD4+, NK cells in a single patient?

Reviewer #3

(Remarks to the Author)

Review of the Manuscript "Understanding the Pathogenesis and Biomarkers of Pediatric Epstein-Barr Virus-Associated Hemophagocytic Lymphohistiocytosis through Single Cell Transcriptomics"

Submitted to: Nature communications

The manuscript presents a thorough proteomics characterization of leukocytes collected in patients with EBV-HLH and compared to healthy donors and patients with self-limiting EBV infection (infective mononucleosis). First, the authors confirm activation of several known pathways in immune cells collected from patients with EBV-HLH, which nicely validates their experimental system as compared to current knowledge. Then they focus on the IDO1+ monocytes and L-kynurenine. The difference in activation of this pathway is subsequently validated on an independent cohort by flow cytometry, metabolomics and direct dosage of tryptophan metabolites. They find that IDO1+ monocytes and L-kynurenine are biomarkers of EBV-HLH disease activity. I compliment the authors on the methodology of the study and how nicely the obtained data is presented. The manuscript is exceptionally well written.

COMMENT 1

I believe that the main limitation of this study is that it is only an association study. It does not provide any proof that mechanisms involving IDO1+ monocytes and the L-kynurenine are causative of any manifestations, but they might be simply an epiphenomenon of the immune system activation. Although some kind of causative mechanism is reasonable to hypothesize, no experiments have been performed to elucidate it. For example, silencing/activating IDO1 or manipulating the L-kynurenine levels in an appropriate cellular or mouse model of EBV-HLH might have been very informative. Without this part, the relevance of this study is limited to identifying potential biomarkers of EBV-HLH and generating hypothesis on pathogenesis. If the authors do not wish to perform further experiments to confirm the pathogenicity as this might be out of the scope of this project, the text should be revised in several points where the authors state to have identified a novel pathogenic mechanism. For example:

> Title: the words "the pathogenesis and" should be removed

> line 61: "The mechanism of action of L-kynurenine in HLH includes the downregulation of NK cell surface activation receptors while promoting the production of numerous pro inflammatory cytokines." This has been only on NK92 cells and cannot be applied directly to the EBV-HLH.

> lines 65-66: "in its pathogenesis, and highlighting their potential as therapeutic targets": before considering IDO1 monocytes and L-kynurenine as therapeutic targets, their role in the causation of EBV-HLH needs to be confirmed. I would suggest removing these words.

> lines 114 "identified a new pathogenic activated" and all further occurrences of the word "pathogenic/pathogenicity" should be carefully evaluated and all claims that pathogenicity of IDO1+ monocytes and L- kynurenine in EBV-HLH has been proven should be removed

> line 317 "Altogether, this highlights the crucial role of monocytes as effector cells in the inflammatory cytokine storm characteristic of HLH.", Although the pathogenic role of monocytes in HLH is well known from other studies, the authors prove here only it is a potential crucial role, not a definitive role.

> lines 482-485: "This finding reveals the important role of L kynurenine in weakening the antiviral function of NK cells and in exacerbating the inflammatory response, thereby playing a new role in the pathological progression of HLH." Should be rephrased, as the authors do not prove that L kynurenine weakens the antiviral function of NK cells (not tested), but only the killing of K562 cells and alteration of the expression of cytotoxicity mediators, cytokines and ROS levels.

> line 488-489: "as key components and biomarkers" I would suggest removing the word component, and keeping only biomarker.

COMMENT 2

The number of patients included in all cohorts (healthy donors, infective mononucleosis, HLH) is small. This is in line with the rarity of EBV-HLH and the difficulty of collecting samples from these very sick patients. However, the limited number of patients included could introduce unforeseen biases in the findings. To mitigate this risk and to allow comparison to future and previous cohorts, a very thorough description of patient characteristics is needed. This will allow to identify patients' factors that might explain potentially differing results in future studies.

I think that the authors should include in the supplementary material the information for every patient in all cohorts at least the information that they show as aggregated in table 1 (i.e. age, EBV-load, etc). Moreover, they should add information for each patient on:

> ethnicity

> when the sample was taken in the course of disease, especially in relation with treatment

> if EBV was found in B, T, or in NK cell

> exact treatment modalities

Finally, normal values for all clinical measurements should be provided where applicable (e.g., ferritin, IL6, and many more)

COMMENT 3

Line 502: "EBV-HLH patients were genetically sequenced before inclusion to confirm the absence of pathogenic gene mutations associated with familial HLH"

Not only genes implicated in familial HLH/cytotoxicity defects (i.e. genes associated with fHLH2-5, Griscelli syndrome, etc) but also genes associated with impaired control of EBV can contribute to EBV-HLH. How exactly were the germline genetic studies performed? Were all the genes implicated in EBV immunity studied, also including the newer ones (e.g. IL27RA, 10.1038/s41586-024-07213-6)? I would suggest adding a supplementary table with a list of investigated genes.

OTHER MINOR COMMENTS

Line 352 "Validation of IDO1+ monocytes and L-kynurenine as new biomarkers for HLH"

Results from paired samples are very interesting. Could the authors confirm that paired samples mean samples taken from the same patient at different timepoints? I would suggest providing details on these timepoints in the supplementary documentation.

Line 337: are these genes associated with type I or type II interferon responses?

Table S1:

What does absolute lymphocyte count upregulation mean? Does it mean increased? Same for downregulation of red blood cells

Line 432-433: "In contrast, reactivating these genes may help restore immune surveillance and antiviral actions of immune cells in EBV-HLH." This claim is not supported by the results

Antonio Marzollo, MD, PhD

Reviewer #4

(Remarks to the Author)

In this manuscript on understanding EBV-HLH using single-cell transcriptomics, the authors utilize single-cell genomics to explore the unique landscape of EBV-HLH, comparing it to EBV-induced IM and healthy pediatric controls. A significant strength of the study is its analysis of a substantial sample size for this rare disease, establishing a robust resource and providing a thorough evaluation of EBV-HLH. However, there is a lack of strong supportive functional data support for the hypotheses presented. Here are some specific major comments:

1. In Fig. 1, specifically Fig. 1e, there is a marked decrease in innate immune cell populations under inflammatory conditions compared to healthy controls, with a more pronounced decrease in cDCs and pDCs than in monocytes. This raises the question: why did the authors focus on monocytes over other innate immune parameters? Given IDO1 established role in dendritic cells and macrophages, the study might have been more informative if it included dendritic cells, which are key antigen-presenting cells, rather than focusing primarily on monocytes. A similar decrease is noted for plasma/B cells, yet the study prioritizes monocytes without potential explanation.

2. How do the authors distinguish whether the decrease in cell populations is a general effect of viral infection or a consequence of uncontrolled inflammation?

3. In the NK cell section, the authors conclude that MAPK leads to cytokine overproduction. Many different pathways are activated in the analysis contributing to this, and if so, why is MAPK given exclusive focus?

4. NF- κ B signaling is consistently upregulated across various cell types in EBV-HLH, with a relatively standardized expression in T and B cells. Since HLH is typically regarded as an adaptive immune-driven disease, it is unclear why the NF- κ B pathway was not experimentally validated in IM versus EBV-HLH patients. Performing this validation would help corroborate the analysis data effectively.

5. In Fig. 4d, monocytes exhibit various cell states; in addition to IDO1 expression in Cluster 6, Cluster 3 (marked by NEAT1) also stands out. This cluster differentiates IM from EBV-HLH through NEAT1 downregulation—a long non-coding RNA

associated with cell proliferation and oxidative stress. Why was this cluster not included in the analysis?

6. In addition to IDO1, did the authors investigate other monocyte markers, such as HLA-DR and CD163 as positive controls when assessing IDO1 through flow cytometry? How does IDO1 expression correlate with other activation markers within both innate and T cell populations? This would help clarify whether IDO1 expression reflects a general activation state at the time of sample collection or if it serves as a distinct biomarker for this condition.

7. To validate IDO1 as an accurate biomarker for EBV-HLH, a monocyte-specific deletion of IDO1 in an HLH model could clarify the effects of IDO1 deficiency on disease pathology better. However, the association between EBV and IDO1 may not be entirely novel; a recent study by Durovic et al. (A metabolic dependency of EBV can be targeted to hinder B cell transformation) demonstrated that an EBV viral protein can induce IDO1 in B cells to facilitate their transformation. Additionally, Durovic et al. observed increased NAD synthesis and mitochondrial pathway activation upon EBV infection, which aligns with the authors' data showing elevated oxidative phosphorylation in their sample sets. Given the specificity of IDO1 to EBV, however, determining if IDO1 reliably distinguishes between EBV-induced IM and EBV-HLH remains challenging.

8. The analysis and major claims seem heavily based on single-cell data without robust experimental validation. The paper primarily characterizes the transcriptional and cell state landscape of EBV-HLH, so the focus should be less on asserting biomarker identification for EBV-HLH.

Other minor comments:

1. Line 53-55: The hypothesis presented in the abstract is misleading. This study is not simply about using single-cell sequencing to compare the transcriptomics of IM and healthy controls, but rather about comparing IM and EBV-HLH to understand the drivers of the transition from IM to HLH.

2. The results section lacks a description of the cohort used in the study and begins directly with single-cell analysis, which may make it challenging for readers from diverse backgrounds to follow the context.

3. Lines 146-147: The statement that EBV-HLH transcriptional signatures are entirely distinct from IM is not accurate. These signatures are both overlapping and distinct, and this should be clarified.

4. Lines 242-243: The claim that aberrant MAPK signaling may lead to cytokine overproduction could benefit from citing studies where MAPK activation is experimentally shown to lead to cytokine overproduction (specifically IFN γ , IL-1 β) relevant to HLH during inflammation or viral infection.

5. Lines 384-385: The effect of kynurenine (KYN) on T and B cells has not been evaluated. Including this analysis would provide additional insights.

6. Line 511: The mention of "tissue processing" is incorrect, as no tissues were used in this study. It should refer specifically to PBMC processing.

7. Instead of a frequency graph (Fig 6a), it would be beneficial to include the gating strategy for monocytes and FACS plots for IDO1 staining.

Reviewer #5

(Remarks to the Author)

Version 1:

Reviewer comments:

Reviewer #1

(Remarks to the Author)

The author has addressed my questions sufficiently, and the manuscript has been revised well.

Reviewer #2

(Remarks to the Author)

The authors performed additional analyses and presented similarities and dissimilarities between patients with different outcomes and different cell-patterns of EBV-infection. Although the whole experiment remains an association study and the differences between the same cell population infected or not-infected by EBV was not provided, the paper is worth publishing as it provides the basis for future molecular analysis of EBV-HLH.

Reviewer #3

(Remarks to the Author)

The authors have fully addressed my concerns.

Reviewer #4

(Remarks to the Author)

The authors have provided a clear and thorough point-by-point rebuttal to the revision requests. In response to our comments, they have added additional data on the involvement of cDCs and pDCs in their cohort, as well as the association of IDO1 with other monocyte activation markers. They have also addressed numerous line-specific edits throughout the manuscript. We acknowledge that including gene knockout mouse models is not always feasible, particularly in studies primarily focused on human samples and clinical observations. We have no further comments, and the revisions appear appropriate for acceptance of the study.

Reviewer #5

(Remarks to the Author)

Reviewer #1 (Remarks to the Author):

Shen et al. performed single-cell RNA sequencing to reveal the immune landscape in pediatric EBV-HLH. They found excessive cytokine secretion by T and NK cells along with a shift in monocyte differentiation towards an inflammatory phenotype. Moreover, L-kynurenine, regulated by IDO1, was specifically elevated in EBV-HLH, highlighting their potential as therapeutic targets. **Overall, this manuscript is well-written and provides new insights into the pathogenesis of EBV-HLH. Specific comments are as follows.**

1. EBV-HLH is related to primary EBV infection in most pediatric cases. Therefore, it is important to distinguish between primary EBV infection and reactivation. Serological data on EBV should be provided. Serological data is also important to discuss a potential weakening of humoral immunity in patients with HLH as stated in the liens 138 – 139. Furthermore, patients with chronic active EBV infection (CAEBV) can develop EBV-HLH. Were CAEBV cases included in this study? If so, the pathogenesis of EBV-HLH related to CAEBV would be different from other EBV-HLH

Response: We have provided detailed clinical information of the samples in **Supplementary Table 1**. The EBV-HLH cases included in this study for single-cell sequencing were confirmed to have no chronic active EBV infection (CAEBV) background.

2. In Figure S2C, the distribution of the T-cell phenotype of HLH5 is similar to that of IM (high proportion of CD8_STMN1). This patient might be classified as IM. Are there any relationships between the distribution of T-cells or other immune cell phenotypes and clinical course?

Response: We have re-evaluated the clinical records of patient HLH5 and confirmed that this individual met the diagnostic criteria for HLH, including persistent fever, hepatosplenomegaly, bone marrow hemophagocytosis, hyperferritinemia, elevated sCD25, and hypofibrinogenemia.

Analysis of the proportional distribution of immune cell subsets revealed considerable heterogeneity across the disease groups. For example, the frequency of

CD8⁺ T cells in IM4, HLH15, and HLH17 was not significantly elevated compared to other samples. Regarding the distribution of internal T cell subsets, HLH6, HLH10, HLH17, IM3, and IM4 partially resembled the healthy volunteer (HV) group in terms of subset proportions. In contrast, HLH16 exhibited similar cellular distribution patterns to HLH11, while HLH7 showed similarities to HLH3 and HLH15.

Importantly, with the exception of HLH7 and HLH16—both of whom succumbed to the disease—all other patients achieved complete remission. These observations suggest that the proportional distribution of immune cell subsets varies among individuals and does not reliably distinguish disease status or predict clinical outcomes. At present, no strong association has been identified between immune cell subset composition and clinical disease course.

3. The effect of L-kynurenine on NK cells is shown in Figure 6. Authors have concluded that L-kynurenine in HLH may downregulate NK cell activation and the production of pro-inflammatory cytokines. On the other hand, pro-inflammatory cytokines can be produced by different immune cell types such as monocytes lymphocytes in EBV-HLH. Does L-kynurenine have the same effect on other immune cells?

Response: To address this, we further investigated the impact of L-kynurenine on additional immune cell types, including monocytic cell lines (THP-1 and U937) and primary human T cells. Our assays revealed that L-kynurenine enhances the production of pro-inflammatory cytokines in both monocytes and T cells, indicating that its effect is not limited to NK cells. These findings suggest that L-kynurenine may act as a pro-inflammatory mediator contributing to the cytokine storm observed in HLH. The corresponding experimental data have been incorporated into **Fig. 6f–g**.

4. The progression of EBV-HLH can differ, and two fatal cases were included in this study. Were there any distinct immunological features in these cases?

Response: The two fatal cases included in our study were HLH7 and HLH16. To explore potential immunological distinctions, we performed a comparative analysis between these two cases (designated as the HLH-D group) and the remaining HLH

cases (HLH-L group).

Within the T cell compartment of the HLH-D group, upregulated genes were predominantly enriched in pathways related to NF- κ B signaling activation, cytokine storm, and intrinsic apoptosis, indicating excessive inflammatory responses and enhanced cell death. In contrast, downregulated genes were enriched in oxidative phosphorylation, positive regulation of leukocyte activation, regulation of hematopoiesis, and positive regulation of binding, suggesting potential impairments in T cell activation and functional responses.

Notably, *ANXA1*, a key endogenous anti-inflammatory mediator, was significantly downregulated in the HLH-D group. Given that *ANXA1* is upregulated in the IM group—presumably to restrain immune overactivation and facilitate disease resolution—its marked downregulation in fatal HLH cases may contribute to uncontrolled inflammation and poor prognosis. These findings support a potential association between *ANXA1* expression and HLH disease severity. Relevant discussion has been added in **lines 260–270**.

In NK cells, we observed that *DUSP1*, a known negative regulator of inflammation, was significantly downregulated in the HLH-D group compared to the HLH-L group. In contrast, *DUSP1* was upregulated in the IM group, further suggesting a protective, anti-inflammatory role. Collectively, these findings indicate that decreased expression of *DUSP1* may be linked to heightened disease severity and worse clinical outcomes in HLH. Corresponding discussion has been incorporated in **lines 332–336**.

5. Single-cell sequencing analysis was performed using a small number of PBMC samples of patients with EBV-HLH in previous reports (Liu et al. PMID: 31914172 and Suzuki et al. 38642164). Are there any similarities or differences in immunological features compared to the previous reports?

Response: The study by Liu et al. (PMID: 31914172) primarily included refractory adult EBV-HLH patients, some of whom harbored genetic defects associated with primary HLH. The study by Suzuki et al. (PMID: 38642164) focused on pediatric patients with EBV-HLH in the context of chronic active EBV infection (CAEBV) and underlying primary HLH-related mutations. In contrast, our cohort comprised 17

pediatric EBV-HLH patients who lacked CAEBV involvement and were genetically confirmed to be free of primary HLH-associated mutations.

All three studies consistently highlight aberrant CD8⁺ T cell responses in EBV-HLH. Similar to previous findings, we observed that CD8⁺ T cells in EBV-HLH patients retained cytotoxic gene expression but exhibited elevated exhaustion markers, suggesting dysfunctional activation.

While Suzuki et al. reported increased monocyte proportions during the acute phase of EBV-HLH and upregulation of type I interferon (IFN-I) pathways in T cells, NK cells, and monocytes, our data did not reveal significant differences in monocyte proportions among healthy volunteers (HV), infectious mononucleosis (IM), and HLH groups. Furthermore, IFN-I pathway scores in HLH samples did not exceed those of controls.

Notably, we identified a significant reduction in plasmacytoid dendritic cells (pDCs)—the major producers of IFN-I—in the HLH group compared to healthy controls, along with downregulation of IFN-I-related gene expression. Moreover, our analysis revealed a unique monocyte developmental trajectory in HLH characterized by an IDO1⁺ monocyte subset that was specifically upregulated in HLH. This subset displayed markedly elevated inflammatory scores, suggesting a potential role in amplifying the cytokine storm associated with disease severity.

These findings underscore both shared and distinct immunological features between our dataset and previous reports, reflecting differences in patient populations and disease context.

6. The raw sequencing data should be accessible to the public.

Response: The raw sequencing data have been deposited in the GSA-Human database: <https://ngdc.cncb.ac.cn/gsa-human/s/X9m197mT>.

Reviewer #2 (Remarks to the Author):

In this study, the authors performed single cell RNA sequencing to compare the transcriptomics of children with IM and EBV-HLH to identify unique immunological characteristics of pediatric EBV-HLH. Patients with EBV-HLH exhibited increased CD8⁺ T cells and decreased plasma cells, with excessive cytokine secretion by T and NK cells. Monocyte differentiation towards an inflammatory phenotype, and the aggregation of IDO1⁺ monocytes were also identified. They also showed that L-kynurenine is selectively increased in sera from EBV-HLH patients and propose IDO and L-kynurenine as possible therapeutic targets. **Overall, the analysis is well done and the study succeeded in providing immunological landscape of EBV-HLH at single cell level.**

However, admitting advantages of the current study in providing detailed information on minor cell populations which are not readily identified by the conventional analyses, the analysis is superficial and lacks depth. Activation of CD8⁺ T cells and increased inflammatory cytokine production by T and NK cells in EBV-HLH has been well described. It has also been reported that activation of RIG-I pathway in monocyte-derived macrophages by viral EBV1 delivered from exosomes derived from EBV-infected cells induces IDO activation. There are many reports regarding the role of IDO in pathogenesis of EBV-associated diseases. Therefore, this paper fails to provide pivotal findings in immunological pathology of EBV-HLH.

Response: We appreciate the reviewer's thoughtful comments and acknowledge the prior literature on immune activation in EBV-associated diseases. While previous studies have reported CD8⁺ T cell activation and inflammatory cytokine production in EBV-HLH, our study provides the most comprehensive single-cell transcriptomic dataset to date, comprising the largest pediatric EBV-HLH cohort analyzed at single-cell resolution.

Importantly, our analysis not only confirms known immune alterations but also identifies novel immunopathological features. These include a distinct monocyte differentiation trajectory, the enrichment of an IDO1⁺ monocyte subset with a high inflammatory score, and the selective elevation of L-kynurenine in EBV-HLH patient sera. Although IDO1 activation in response to EBV has been described, the

identification of a functionally distinct IDO1⁺ monocyte population within the peripheral blood of EBV-HLH patients has not been previously reported. Furthermore, our data highlight L-kynurenine as an active participant in immune dysregulation beyond NK cells, extending its relevance to monocytes and T cells.

Thus, our study not only validates previous findings but also provides new mechanistic insights into the immunopathogenesis of EBV-HLH, underscoring the therapeutic potential of targeting the IDO1–kynurenine axis in this disease context.

Major concerns:

1. Diagnostic criteria of IM and EBV-HLH for this study is not provided. EBV-infected cell populations in each EBV-HLH patient should be provided. ◦

Response: We have now included the diagnostic criteria for both infectious mononucleosis (IM) and EBV-HLH in **lines 126-131**. Additionally, we have updated the clinical information for the EBV-HLH patients, including detailed results on the EBV-infected cell populations, as shown in the revised supplementary materials.

2. Detailed immunophenotypic analysis of each patient depending on the disease severity and infected cell types should be performed.

Response: We thank the reviewer for this important suggestion. To address the relationship between immunophenotypes, disease severity, and EBV-infected cell types, we performed stratified analyses based on clinical outcomes and EBV infection status.

1) Disease severity (fatal vs. non-fatal cases):

Among the HLH patients, HLH7 and HLH16 represented the two fatal cases (designated as the HLH-D group), while the remaining patients formed the HLH-L group. Comparative analysis revealed that T cells from the HLH-D group showed upregulation of genes involved in NF- κ B signaling, cytokine storm, and intrinsic apoptosis, consistent with uncontrolled inflammation and cell death. In contrast, genes associated with oxidative phosphorylation, hematopoiesis, and leukocyte activation were downregulated, suggesting impaired immune function. Notably, *ANXA1*, an anti-inflammatory mediator, was markedly downregulated in the HLH-D group, implicating its potential association with disease severity and prognosis. In NK cells, *DUSP1*, a

negative regulator of inflammation, was also significantly reduced in the HLH-D group but upregulated in the IM group, further supporting its relevance to disease outcomes. These findings are discussed in detail in **lines 260–270 and 332–336**.

2) EBV-infected T cells (EBV-T vs. non-EBV-T):

Patients were classified into EBV-T and non-EBV-T groups based on clinical detection of EBV in T cells. The EBV-T group exhibited enhanced expression of innate immune-related genes (e.g., type I interferon response, viral response, and NF- κ B signaling), alongside increased leukocyte migration and cytokine production, indicating heightened systemic inflammation. However, genes associated with adaptive immunity—such as T cell activation, cytotoxicity, and antigen presentation—were downregulated, suggesting impaired effector function. The co-occurrence of innate immune activation and adaptive suppression may underlie the cytokine storm in EBV-T patients. Additionally, HLH-related genes including *SH2D1A*, *ADA*, *RAB27A*, and *SLAMF7* were significantly downregulated in EBV-T patients compared to IM and non-EBV-T groups, implicating EBV infection in suppressing pathways critical for immune regulation and viral clearance. Furthermore, *CD81*, a molecule involved in immune synapse formation, was progressively reduced from IM to non-EBV-T to EBV-T groups, suggesting a shared immunopathogenic mechanism in HLH regardless of infection status. These data are discussed in **lines 232–259**.

3) EBV-infected NK cells (EBV-NK vs. non-EBV-NK):

Based on EBV detection in NK cells, we stratified patients into EBV-NK and non-EBV-NK groups. NK cells from the EBV-NK group displayed metabolic reprogramming, with upregulated genes involved in ATP metabolism and oxidative phosphorylation, supporting elevated energy demands during antiviral responses. However, despite enhanced cytokine production and cytotoxic capacity, these cells concurrently exhibited immunosuppressive features, including upregulation of apoptosis and negative regulation of cytotoxicity, suggesting immune exhaustion. Downregulated genes in EBV-NK cells were enriched in antigen presentation, lymphocyte differentiation, and cytotoxicity, highlighting impaired immune

surveillance. These defects may hinder T cell-mediated clearance of EBV-infected NK cells and facilitate viral persistence. Furthermore, *LYST* and *DUSP1*, key regulators of HLH pathogenesis and inflammation, were significantly downregulated in EBV-NK but not in non-EBV-NK, IM, or HV groups, indicating direct modulation by EBV infection. Relevant discussions are provided in **lines 313–336**.

Together, these analyses demonstrate that both disease severity and EBV-infected cell types are associated with distinct immunological signatures, supporting the heterogeneity of EBV-HLH and providing mechanistic insight into its pathogenesis.

3. It is more interesting and important to know the differences between specific cell populations infected or not-infected by EBV. In other words, what differences can be found in EBV-infected and EBV-uninfected CD8+, CD4+, NK cells in a single patient?

Response: We appreciate the reviewer’s insightful question. To address this, we attempted to identify EBV-infected cells by aligning single-cell RNA-seq reads to the EBV reference genome (Liu et al., PMID: 33531485). However, likely due to technical limitations such as low viral transcript abundance and insufficient sequencing depth, we were unable to reliably detect EBV-positive cells in the current dataset.

We fully acknowledge the importance of dissecting the immunological differences between EBV-infected and uninfected CD8+, CD4+, and NK cells within individual patients. In future studies, we plan to utilize more sensitive, virus-targeted single-cell approaches—such as hybrid capture or enrichment strategies—to enhance EBV transcript detection and enable precise characterization of infected versus uninfected immune cell subsets.

Reviewer #3 (Remarks to the Author):

The manuscript presents a thorough proteomics characterization of leukocytes collected in patients with EBV-HLH and compared to healthy donors and patients with self-limiting EBV infection (infective mononucleosis). First, the authors confirm activation of several known pathways in immune cells collected from patients with EBV-HLH, which nicely validates their experimental system as compared to current knowledge. Then they focus on the IDO1+ monocytes and L-kynurenine. The difference in activation of this pathway is subsequently validated on an independent cohort by flow cytometry, metabolomics and direct dosage of tryptophan metabolites. They find that IDO1+ monocytes and L-kynurenine are biomarkers of EBV-HLH disease activity. **I compliment the authors on the methodology of the study and how nicely the obtained data is presented. The manuscript is exceptionally well written.**

COMMENT 1

I believe that the main limitation of this study is that it is only an association study. It does not provide any proof that mechanisms involving IDO1+ monocytes and the L-kynurenine are causative of any manifestations, but they might be simply an epiphenomenon of the immune system activation. Although some kind of causative mechanism is reasonable to hypothesize, no experiments have been performed to elucidate it. For example, silencing/activating IDO1 or manipulating the L-kynurenine levels in an appropriate cellular or mouse model of EBV-HLH might have been very informative. Without this part, the relevance of this study is limited to identifying potential biomarkers of EBV-HLH and generating hypothesis on pathogenesis. If the authors do not wish to perform further experiments to confirm the pathogenicity as this might be out of the scope of this project, the text should be revised in several points where the authors state to have identified a novel pathogenic mechanism. For example:

- > Title: the words “the pathogenesis and” should be removed.

Response: We have deleted the words “the pathogenesis and” in the title.

- > line 61: “ The mechanism of action of L-kynurenine in HLH includes the downregulation of NK cell surface activation receptors while promoting the production

of numerous pro inflammatory cytokines.” This has been only on NK92 cells and cannot be applied directly to the EBV-HLH.

Response: Per your suggestion, we have revised lines 61–62 to read: 'The mechanism of action of L-kynurenine in HLH involves the induction of multiple pro-inflammatory cytokines'.

> lines 65-66: “in its pathogenesis, and highlighting their potential as therapeutic targets” : before considering IDO1 monocytes and L-kynurenine as therapeutic targets, their role in the causation of EBV-HLH needs to be confirmed. I would suggest removing these words.

Response: The relevant statements have been removed as suggested.

> lines 114 “identified a new pathogenic activated” and all further occurrences of the word “pathogenic/pathogenicity” should be carefully evaluated and all claims that pathogenicity of IDO1+ monocytes and L- kynurenine in EBV-HLH has been proven should be removed.

Response: The relevant statements have been removed.

> line 317 “Altogether, this highlights the crucial role of monocytes as effector cells in the inflammatory cytokine storm characteristic of HLH.” , Although the pathogenic role of monocytes in HLH is well known from other studies, the authors prove here only it is a potential crucial role, not a definitive role.

Response: We have revised lines 470–471 as follows: 'Collectively, these findings indicate that monocytes may function as effector cells and contribute significantly to the inflammatory cytokine storm observed in HLH.'

> lines 482-485: “This finding reveals the important role of L kynurenine in weakening the antiviral function of NK cells and in exacerbating the inflammatory response, thereby playing a new role in the pathological progression of HLH.” Should be rephrased, as the authors do not prove that L kynurenine weakens the antiviral function of NK cells (not tested), but only the killing of K562 cells and alteration of the

expression of cytotoxicity mediators, cytokines and ROS levels.

Response: Based on your suggestion, we focused subsequent experiments on the pro-inflammatory effects of KYN, and therefore removed the data related to NK cell cytotoxicity and ROS production.

> line 488-489: “as key components and biomarkers” I would suggest removing the word component, and keeping only biomarker.

Response: We have revised **lines 639–640** as suggested: 'Elevated IDO1 expression and kynurenine levels may serve as potential auxiliary biomarkers for distinguishing EBV-HLH.'

In addition, we conducted additional in vitro experiments, which confirmed that overexpression of IDO1 in monocytic cell lines (THP1 and U937) induces a pronounced inflammatory phenotype. Moreover, exogenous kynurenine supplementation markedly increased the expression of pro-inflammatory cytokines in monocytic cell lines, NK cell lines, and primary T cells. These findings support the role of IDO1 and kynurenine as pro-inflammatory mediators contributing to the cytokine storm. The detailed results have been presented in **Fig. 6f-g**. We sincerely appreciate your insightful suggestions and have carefully revised the discussion to enhance clarity, precision, and scientific rigor.

COMMENT 2

The number of patients included in all cohorts (healthy donors, infective mononucleosis, HLH) is small. This is in line with the rarity of EBV-HLH and the difficulty of collecting samples from these very sick patients. However, the limited number of patients included could introduce unforeseen biases in the findings. To mitigate this risk and to allow comparison to future and previous cohorts, a very thorough description of patient characteristics is needed. This will allow to identify patients' factors that might explain potentially differing results in future studies.

I think that the authors should include in the supplementary material the information for every patient in all cohorts at least the information that they show as aggregated in

table 1 (i.e. age, EBV-load, etc). Moreover, they should add information for each patient on:

> ethnicity

> when the sample was taken in the course of disease, especially in relation with treatment

> if EBV was found in B, T, or in NK cell

> exact treatment modalities

Finally, normal values for all clinical measurements should be provided where applicable (e.g., ferritin, IL6, and many more)

Response: We appreciate the reviewer’s insightful comment regarding the limited sample size and the potential for bias. As correctly noted, EBV-HLH is a rare and severe condition, which poses substantial challenges for sample collection. To address this concern and facilitate comparison with previous and future studies, we have provided more granular clinical information for each patient in **Supplementary Table 1**. This includes individual data previously presented in aggregate in Table 1 (e.g., age and EBV load), as well as the additional variables requested: patient ethnicity, timing of sample collection in relation to disease course and treatment initiation, EBV-infected cell type (B, T, or NK cells), and detailed treatment regimens. Furthermore, we have included reference (normal) ranges for all clinical measurements where applicable. We hope these additions will improve the transparency and reproducibility of our findings and provide a valuable resource for future investigations.

COMMENT 3

Line 502: “EBV-HLH patients were genetically sequenced before inclusion to confirm the absence of pathogenic gene mutations associated with familial HLH”

Not only genes implicated in familial HLH/cytotoxicity defects (i.e. genes associated with fHLH2-5, Griscelli syndrome, etc) but also genes associated with impaired control of EBV can contribute to EBV-HLH. How exactly were the germline genetic studies performed? Were all the genes implicated in EBV immunity studied, also including the newer ones (e.g. IL27RA, 10.1038/s41586-024-07213-6)? I would suggest adding a supplementary table with a list of investigated genes.

Response: All EBV-HLH samples included in the single-cell sequencing underwent whole-exome sequencing to exclude genetic variants. We further verified the genetic sequencing results to ensure the absence of mutations in HLH-related and primary immunodeficiency-associated genes. The relevant gene panel is provided in **Supplementary Table 2**.

Other minor comments:

Line 352 “Validation of IDO1+ monocytes and L-kynurenine as new biomarkers for HLH” Results from paired samples are very interesting. Could the authors confirm that paired samples mean samples taken from the same patient at different timepoints? I would suggest providing details on these timepoints in the supplementary documentation.

Response: Yes, the paired samples in the validation cohort were collected from the same patients at two different timepoints—prior to and following treatment. We have now included the timepoints of sample collection in **lines 511-513** as suggested: 'The samples of HLH-T group were collected within 4-6 weeks after standard treatment. The remaining samples were collected within 0-3 days after admission (acute phase, prior to treatment).'

Line 337: are these genes associated with type I or type II interferon responses?

Response: *IFI27*, *IRF7*, *IFITM1*, *ISG15*, and *IRF1* are primarily associated with the type I interferon response, whereas *GBP1* and *IFNGR2* are linked to the type II interferon response.

Table S1: What does absolute lymphocyte count upregulation mean? Does it mean increased? Same for downregulation of red blood cells

Response: The terms "upregulated" and "downregulated" here specifically refer to an increase or decrease in cell numbers, respectively. To avoid ambiguity, we have revised the descriptions in the table accordingly.

Line 432-433: “In contrast, reactivating these genes may help restore immune

surveillance and antiviral actions of immune cells in EBV-HLH.” This claim is not supported by the results

Response: To improve accuracy, we have rephrased **lines 587–589** as follows: ‘Targeted restoration of their function could theoretically enhance immune surveillance against EBV, which may represent a potential therapeutic strategy.’

Reviewer #4 (Remarks to the Author):

In this manuscript on understanding EBV-HLH using single-cell transcriptomics, the authors utilize single-cell genomics to explore the unique landscape of EBV-HLH, comparing it to EBV-induced IM and healthy pediatric controls. **A significant strength of the study is its analysis of a substantial sample size for this rare disease, establishing a robust resource and providing a thorough evaluation of EBV-HLH.** However, there is a lack of strong supportive functional data support for the hypotheses presented. Here are some specific major comments:

1. In Fig. 1, specifically Fig. 1e, there is a marked decrease in innate immune cell populations under inflammatory conditions compared to healthy controls, with a more pronounced decrease in cDCs and pDCs than in monocytes. This raises the question: why did the authors focus on monocytes over other innate immune parameters? Given IDO1 established role in dendritic cells and macrophages, the study might have been more informative if it included dendritic cells, which are key antigen-presenting cells, rather than focusing primarily on monocytes. A similar decrease is noted for plasma/B cells, yet the study prioritizes monocytes without potential explanation.

Response: Thank you for this important comment. In our initial analysis, we focused on monocytes due to their markedly elevated inflammatory scores and their known contribution to the cytokine storm in HLH. This led to the identification of a disease-specific IDO1⁺ inflammatory monocyte subset, which we prioritized for in-depth characterization.

However, we agree that dendritic cells (DCs), particularly cDCs and pDCs, are critical antigen-presenting cells and may also play important roles in HLH pathogenesis. In response to your suggestion, we performed additional analyses and have now included detailed immunological profiling of cDCs and pDCs in the revised manuscript (lines 388–422).

Briefly, both cDCs and pDCs were significantly reduced in HLH. Residual cDCs showed increased expression of genes involved in antiviral responses, MHC class I presentation, and CD8⁺ T cell activation, suggesting a hyperactivated state. However, concurrent downregulation of MHC class II and ribosome-related pathways implies

impaired CD4⁺ T cell activation and cellular exhaustion. pDCs were also depleted, with HLH-associated pDCs displaying enriched MHC class I antigen presentation and oxidative phosphorylation pathways, indicating immunologic activation under metabolic stress. Downregulation of IFN- β production and myeloid differentiation-related genes may contribute to pDC depletion and impaired antiviral immunity.

These findings suggest that DCs in HLH contribute to CD8⁺ T cell hyperactivation while undergoing functional exhaustion. We have incorporated these results into the revised manuscript and Supplementary Figures, as well as expanded the discussion to reflect their relevance to HLH pathogenesis.

2. How do the authors distinguish whether the decrease in cell populations is a general effect of viral infection or a consequence of uncontrolled inflammation?

Response: Thank you for raising this insightful question. Distinguishing whether the observed reduction in immune cell populations is primarily due to direct viral effects or secondary to uncontrolled inflammation is indeed complex. To date, there remains a paucity of conclusive literature that systematically dissects the relative contributions of direct viral cytopathic effects versus inflammation-mediated bystander damage in driving the observed depletion of immune cells. This knowledge gap represents a compelling research direction, as elucidating the predominant mechanism holds significant implications for understanding viral pathogenesis and developing targeted immunomodulatory interventions. In theory, longitudinal monitoring of viral infection, cytokine levels, treatment responses and immune cell dynamics could help delineate these mechanisms. For example, persistent high intracellular viral load accompanied by immune cell depletion would support a direct viral cytopathic effect, whereas a stable or declining viral load with escalating cytokine levels and immune cell loss may suggest inflammation-driven injury. Furthermore, a significant recovery in immune cells following antiviral therapy may suggest that virus-induced immunosuppression through direct infection plays a dominant role. Conversely, if immunomodulatory treatments (e.g., glucocorticoids or cytokine antagonists) effectively reverse cytopenia and alleviate clinical symptoms, this strongly supports a secondary immune injury mechanism driven by dysregulated hyperinflammation.

However, in practice, these mechanisms likely act in concert, with direct viral effects predominating in early stages of infection and inflammatory injury contributing more significantly during disease progression. As our current dataset is limited to single time-point analyses, we are unable to assess these temporal dynamics directly. However, this critical question warrants further investigation in our subsequent studies.

3. In the NK cell section, the authors conclude that MAPK leads to cytokine overproduction. Many different pathways are activated in the analysis contributing to this, and if so, why is MAPK given exclusive focus?

Response: While our pathway analysis revealed the activation of multiple signaling cascades in NK cells, we specifically focused on the MAPK pathway due to the distinct and consistent expression pattern of *DUSP1*, a critical negative regulator of MAPK signaling. Notably, *DUSP1* was significantly upregulated in the IM group compared to healthy volunteers (HV), but markedly downregulated in the HLH group (Fig. 3k). Other genes involved in the negative regulation of the MAPK cascade, including *DUSP2*, *NCOR1*, *STK38*, *PDCD4*, and *ITCH*, were also downregulated in the HLH group. Furthermore, within the HLH cohort, *DUSP1* expression was significantly lower in patients with poor prognosis (HLH-D) compared to those with favorable outcomes (HLH-L) (Fig. 3l).

This opposing regulation pattern between disease groups and its correlation with disease severity suggest that impaired *DUSP1*-mediated regulation of MAPK signaling may contribute to uncontrolled cytokine production in HLH. Therefore, although other pathways are also activated, we emphasized the MAPK pathway due to the prognostic relevance and functional implication of *DUSP1* in modulating inflammatory responses in NK cells.

4. NF- κ B signaling is consistently upregulated across various cell types in EBV-HLH, with a relatively standardized expression in T and B cells. Since HLH is typically regarded as an adaptive immune-driven disease, it is unclear why the NF- κ B pathway was not experimentally validated in IM versus EBV-HLH patients. Performing this validation would help corroborate the analysis data effectively.

Response: To experimentally validate the transcriptomic findings, we measured phosphorylated p65 (a key component of canonical NF- κ B signaling) in T cells, NK cells, B cells, and monocytes from both IM and HLH patients. The results, shown in **Fig. 2h, Fig. 3g, Fig. 4j, and Fig. S3i**, demonstrate significantly higher levels of phosphorylated p65 in all examined immune cell types from HLH patients compared to those from IM patients. These findings confirm a broad and robust activation of the canonical NF- κ B pathway in EBV-HLH, supporting the scRNA-seq-based inference and underscoring its relevance to the pathogenesis of the disease.

5. In Fig. 4d, monocytes exhibit various cell states; in addition to IDO1 expression in Cluster 6, Cluster 3 (marked by NEAT1) also stands out. This cluster differentiates IM from EBV-HLH through NEAT1 downregulation — a long non-coding RNA associated with cell proliferation and oxidative stress. Why was this cluster not included in the analysis?

Response: **Fig. 4d** presents the Ro/e analysis of monocyte clusters, which may be influenced by sample variability. While Cluster 3, marked by NEAT1, appeared notable, further analysis showed that its proportion did not significantly differ between the IM and HV groups. Therefore, we did not prioritize it for detailed follow-up.

However, in response to your comment and interest in monocyte subsets enriched in IM, we performed a focused analysis of Cluster 8, which was significantly elevated in the IM group but not in HLH. Functional enrichment of Cluster 8 revealed upregulation of genes involved in MHC class II antigen presentation (e.g., “antigen processing and presentation of exogenous peptide antigen” and “MHC class II protein complex assembly”), suggesting a potential role in supporting T cell activation. In contrast, downregulated genes were associated with humoral responses, cytokine production, and inflammatory pathways, indicating a restrained inflammatory profile.

These findings suggest that Cluster 8 represents a monocyte subset in IM that differentiates toward an antigen-presenting, immunoregulatory phenotype—possibly contributing to the self-limiting nature of IM. Conversely, monocytes in HLH adopt a pro-inflammatory state, contributing to the cytokine storm. These results have been added to the revised manuscript (**lines 434–445**) to clarify the differential monocyte

states in EBV-HLH versus IM.

6. In addition to IDO1, did the authors investigate other monocyte markers, such as HLA-DR and CD163 as positive controls when assessing IDO1 through flow cytometry? How does IDO1 expression correlate with other activation markers within both innate and T cell populations? This would help clarify whether IDO1 expression reflects a general activation state at the time of sample collection or if it serves as a distinct biomarker for this condition.

Response: To evaluate whether IDO1 expression reflects a general activation state or represents a distinct biomarker for HLH, we analyzed its association with established monocyte activation markers—HLA-DR, CD163, and CX3CR1—using flow cytometry in an independent validation cohort comprising IM and EBV-HLH patients.

Our results demonstrated a significantly higher proportion of IDO1⁺ monocytes in the HLH group compared to IM. In contrast, HLA-DR was uniformly expressed at high levels in both groups, with no significant difference in the proportion of HLA-DR⁺ cells. Similarly, the proportions of CD163⁺ and CX3CR1⁺ monocytes did not differ significantly between HLH and IM.

Further stratification within the HLH group showed no significant difference in HLA-DR⁺, CD163⁺, or CX3CR1⁺ populations between IDO1⁺ and IDO1⁻ monocytes. Additionally, the frequency of IDO1⁺ cells remained consistent across subgroups defined by the expression of these other markers.

These findings suggest that IDO1 upregulation in HLH is not simply a reflection of a general monocyte activation state but may represent a distinct, disease-specific inflammatory signature. The relevant discussion has been added to **lines 523–535** of the revised manuscript.

7. To validate IDO1 as an accurate biomarker for EBV-HLH, a monocyte-specific deletion of IDO1 in an HLH model could clarify the effects of IDO1 deficiency on disease pathology better. However, the association between EBV and IDO1 may not be entirely novel; a recent study by Durovic et al. (A metabolic dependency of EBV can be targeted to hinder B cell transformation) demonstrated that an EBV viral protein can induce IDO1 in B cells to facilitate their transformation. Additionally, Durovic et al. observed increased NAD synthesis and mitochondrial pathway activation upon EBV

infection, which aligns with the authors' data showing elevated oxidative phosphorylation in their sample sets. Given the specificity of IDO1 to EBV, however, determining if IDO1 reliably distinguishes between EBV-induced IM and EBV-HLH remains challenging.

Response: We fully agree that the use of monocyte-specific IDO1 knockout models would be invaluable for mechanistic validation of IDO1's contribution to HLH pathogenesis. However, as this study is based on patient-derived clinical samples, *in vivo* functional models were beyond the scope of the current work.

Nonetheless, through integrated clinical analyses—including flow cytometry, metabolomics, and functional assays—we obtained several key findings:

1) Specificity of IDO1 expression: IDO1 expression in monocytes was significantly elevated in EBV-HLH patients compared to both EBV-IM patients and healthy controls, with no statistical difference observed between the IM and control groups (**Fig. 6a**). This suggests that IDO1 expression is not a general marker of EBV infection but is specifically associated with the HLH disease state.

2) Functional relevance: *In vitro* overexpression of IDO1 in monocytic cell lines (THP1 and U937) induced a pronounced inflammatory phenotype. Moreover, exogenous kynurenine supplementation enhanced pro-inflammatory cytokine expression across monocytic cell lines, NK cells, and primary T cells. These results (presented in **Fig. 6f-g**) support the hypothesis that the IDO1–kynurenine axis contributes to immune dysregulation in HLH through a metabolite-driven mechanism.

While we acknowledge that the link between EBV and IDO1 induction is not entirely novel—as highlighted by the study by Durovic et al., which demonstrated IDO1 upregulation in EBV-transformed B cells—we believe our findings offer new insight. Specifically, we show that IDO1 expression in monocytes, rather than B cells, can serve as a discriminative marker for EBV-HLH versus EBV-IM, thereby expanding our understanding of IDO1's role in EBV-associated immune pathology.

We have revised the Discussion section accordingly to incorporate a more nuanced view of IDO1's diagnostic potential, including a citation to Durovic et al. to contextualize our findings within the broader literature.

8. The analysis and major claims seem heavily based on single-cell data without robust experimental validation. The paper primarily characterizes the transcriptional and cell state landscape of EBV-HLH, so the focus should be less on asserting biomarker identification for EBV-HLH.

Response: Given the rarity and clinical severity of EBV-HLH, this study primarily focused on clinical sample-based analyses rather than extensive experimental investigations. As such, our work is intended to serve as an exploratory and hypothesis-generating resource.

In response to your suggestion, we have revised the manuscript to more accurately reflect this scope. Specifically:

The Abstract and Discussion sections have been updated to emphasize the primary objective of our study as a systematic characterization of the immunometabolic microenvironment in EBV-HLH, rather than asserting definitive biomarker identification.

A new statement has been added (**lines 115–120**): “Additionally, L-kynurenine, a product regulated by IDO1, may serve as a pro-inflammatory factor in EBV-HLH. Collectively, our study provides a single-cell transcriptomic resource for understanding pediatric EBV-HLH pathogenesis and uncovers a potential link between metabolic reprogramming and inflammatory storm, offering a multi-layered target framework for future mechanistic investigations.”

Assertive language, such as “IDO1 as a diagnostic biomarker for EBV-HLH” (**lines 639–642**), has been revised to a more cautious interpretation: “Elevated IDO1 expression and kynurenine levels may serve as potential auxiliary indicators for distinguishing EBV-HLH. However, the biomarker validity requires further confirmation through prospective multicenter cohorts and functional experiments.”

These changes aim to appropriately position our findings as foundational for future experimental validation while maintaining scientific rigor and transparency.

Other minor comments:

1.Line 53-55: The hypothesis presented in the abstract is misleading. This study is not simply about using single-cell sequencing to compare the transcriptomics of IM and

healthy controls, but rather about comparing IM and EBV-HLH to understand the drivers of the transition from IM to HLH.

Response: To enhance clarity, we have revised **lines 54-56** as follows: ‘In this study, by enrolling children with IM and healthy volunteers as controls, we utilized single-cell RNA sequencing to identify unique immunological characteristics of EBV-HLH.’

2. The results section lacks a description of the cohort used in the study and begins directly with single-cell analysis, which may make it challenging for readers from diverse backgrounds to follow the context.

Response: To improve clarity and provide better context for readers, we have added a detailed description of the study cohort in **lines 123–145** of the revised manuscript. This addition outlines the clinical characteristics, grouping criteria, and sample types used in our analysis, thereby enhancing the overall accessibility and interpretability of the results.

3. Lines 146-147: The statement that EBV-HLH transcriptional signatures are entirely distinct from IM is not accurate. These signatures are both overlapping and distinct, and this should be clarified.

Response: We agree that the original statement was too definitive. To address this, we have revised **lines 172–174** to read: “Overall, this finding suggests that EBV-HLH exhibits a unique transcriptional profile, characterized by both shared and distinct patterns of immune cell dysfunction when compared to IM and healthy control groups.” This revision more accurately reflects the observed data and acknowledges the complexity of the transcriptional landscape.

4. Lines 242-243: The claim that aberrant MAPK signaling may lead to cytokine overproduction could benefit from citing studies where MAPK activation is experimentally shown to lead to cytokine overproduction (specifically IFN γ , IL-1 β) relevant to HLH during inflammation or viral infection.

Response: In response, we have expanded the background discussion in **lines 337–348** to provide additional context on the roles of the NF- κ B and MAPK signaling pathways in regulating pro-inflammatory cytokine production, particularly in the context of inflammation and viral infection relevant to HLH. We have also included appropriate citations to support the link between MAPK activation and cytokine

overproduction.

5.Lines 384-385: The effect of kynurenine (KYN) on T and B cells has not been evaluated. Including this analysis would provide additional insights.

Response: In response, we further evaluated the pro-inflammatory effects of kynurenine in vitro using monocytic cell lines (THP1 and U937) as well as primary T cells. The results demonstrated that exogenous kynurenine significantly enhanced pro-inflammatory cytokine expression, supporting its role as a potential driver of hyperinflammation in HLH. These findings have been incorporated into **Fig. 6f–g** of the revised manuscript.

6.Line 511: The mention of "tissue processing" is incorrect, as no tissues were used in this study. It should refer specifically to PBMC processing.

Response: We have corrected it.

7. Instead of a frequency graph (Fig 6a), it would be beneficial to include the gating strategy for monocytes and FACS plots for IDO1 staining.

Response: We have now included the gating strategy for monocyte identification and representative flow cytometry plots for IDO1 staining in **Fig. S7** of the revised manuscript to enhance transparency and reproducibility of the analysis.